# Rabies virus-based barcoded neuroanatomy resolved by single-cell RNA and in situ sequencing

**Aixin Zhang[1†], Lei Jin[2†‡], Shenqin Yao[1], Makoto Matsuyama[2§], Cindy TJ van Velthoven[1], Heather Anne Sullivan[2], Na Sun[3,4], Manolis Kellis[3,4], Bosiljka Tasic[1], Ian Wickersham[2]\*, Xiaoyin Chen[1]\***

[1]Allen Institute for Brain Science, Seattle, United States; [2]McGovern Institute for Brain Research, Massachusetts Institute of Technology, Cambridge, United States; [3]Department of Electrical Engineering and Computer Science, Massachusetts Institute of Technology, Broad Institute of MIT and Harvard, Cambridge, United States; [4]Broad Institute of MIT and Harvard, Cambridge, United States

**\*For correspondence:**
wickersham@mit.edu (IW);
xiaoyin.chen@alleninstitute.org
(XC)

[†]These authors contributed equally to this work

**Present address:** [‡]Lingang Laboratory, Shanghai, China; [§]Metcela Inc, Kanagawa, Japan

**Abstract** Mapping the connectivity of diverse neuronal types provides the foundation for understanding the structure and function of neural circuits. High-throughput and low-cost neuroanatomical techniques based on RNA barcode sequencing have the potential to map circuits at cellular resolution and a brain-wide scale, but existing Sindbis virus-based techniques can only map long-range projections using anterograde tracing approaches. Rabies virus can complement anterograde tracing approaches by enabling either retrograde labeling of projection neurons or monosynaptic tracing of direct inputs to genetically targeted postsynaptic neurons. However, barcoded rabies virus has so far been only used to map non-neuronal cellular interactions in vivo and synaptic connectivity of cultured neurons. Here we combine barcoded rabies virus with single-cell and in situ sequencing to perform retrograde labeling and transsynaptic labeling in the mouse brain. We sequenced 96 retrogradely labeled cells and 295 transsynaptically labeled cells using single-cell RNA-seq, and 4130 retrogradely labeled cells and 2914 transsynaptically labeled cells in situ. We found that the transcriptomic identities of rabies virus-infected cells can be robustly identified using both single-cell RNA-seq and in situ sequencing. By associating gene expression with connectivity inferred from barcode sequencing, we distinguished long-range projecting cortical cell types from multiple cortical areas and identified cell types with converging or diverging synaptic connectivity. Combining in situ sequencing with barcoded rabies virus complements existing sequencing-based neuroanatomical techniques and provides a potential path for mapping synaptic connectivity of neuronal types at scale.

## eLife assessment

This study presents an **important** tool for tracking the connectivity of neurons in mouse and potentially other mammals using a combined approach of barcoded rabies virus libraries and spatial transcriptomics. The data supporting the technique are **convincing**, the validation against known anatomical knowledge is rigorous, and the authors advance the techniques by combing them in vivo. Overall, this is a very good paper describing a technique for tracking neural circuits.

**eLife digest** In the brain, messages are relayed from one cell to the next through intricate networks of axons and dendrites that physically interact at junctions known as synapses. Mapping out this synaptic connectivity – that is, exactly which neurons are connected via synapses – remains a major challenge.

Monosynaptic tracing is a powerful approach that allows neuroscientists to explore neural networks by harnessing viruses which spread between neurons via synapses. and in particular the rabies virus. This pathogen travels exclusively from 'postsynaptic' to 'presynaptic' neurons – from the cell that receives a message at a synapse, back to the one that sends it. A modified variant of the rabies virus can therefore be used to reveal the presynaptic cells connecting to a population of neurons in which it has been originally introduced. However, this method does not allow scientists to identify the exact postsynaptic neuron that each presynaptic cell is connected to.

One way to bypass this issue is to combine monosynaptic tracing with RNA barcoding to create distinct versions of the modified rabies virus, which are then introduced into separate populations of neurons. Tracking the spread of each version allows neuroscientists to spot exactly which presynaptic cells signal to each postsynaptic neuron. So far, this approach has been used to examine synaptic connectivity in neurons grown in the laboratory, but it remains difficult to apply it to neurons in the brain.

In response, Zhang, Jin et al. aimed to demonstrate how monosynaptic tracing that relies on barcoded rabies viruses could be used to dissect neural networks in the mouse brain. First, they confirmed that it was possible to accurately detect which version of the virus had spread to presynaptic neurons using both in situ and single-cell RNA sequencing. Next, they described how this information could be analysed to build models of potential neural networks, and what type of additional experiments are required for this work. Finally, they used the approach to identify neurons that tend to connect to the same postsynaptic cells and then investigated what these have in common, showing how the technique enables a finer understanding of neural circuits.

Overall, the work by Zhang, Jin et al. provides a comprehensive review of the requirements and limitations associated with monosynaptic tracing experiments based on barcoded rabies viruses, as well as how the approach could be optimized in the future. This information will be of broad interest to scientists interested in mapping neural networks in the brain.

## Introduction

The connectivity of diverse types of neurons constrains the flow and processing of information in neural circuits. Neuroanatomical techniques based on microscopy rely on sparse labeling combined with optical tracing to achieve single-cell resolution (*Winnubst et al., 2019*; *Peng et al., 2021*; *Gao et al., 2022*). However, these approaches are usually labor-intensive and difficult to scale up. Furthermore, because many of these approaches rely on specialized imaging platforms, they are difficult to combine with transcriptomic interrogation for defining cell types. Thus, developing high-throughput neuroanatomical techniques that can map neuronal connectivity and associate connectivity with gene expression at single-cell resolution could help generate new insights into the organization of neural circuits that are difficult to gain using conventional neuroanatomical techniques.

RNA barcoding-based neuroanatomical techniques are a new approach that dramatically improves the throughput of single-neuron projection mapping (*Kebschull et al., 2016*; *Chen et al., 2019*; *Sun et al., 2021*). In these techniques, Sindbis virus is used to express random RNA sequences, or RNA 'barcodes', to uniquely label each neuron. The barcoded virus RNAs replicate and distribute throughout the somata and axons; therefore, matching barcodes in the axons to those in the somata reveal the axonal projections of individual neurons. Because many barcode molecules can be sequenced in parallel, barcoding-based neuroanatomical approaches can determine the long-range projections of tens of thousands of neurons in single animals. These barcoding-based techniques have been applied to various brain regions (*Kebschull et al., 2016*; *Han et al., 2018*; *Chen et al., 2019*; *Gergues et al., 2020*; *Klingler et al., 2021*; *Mathis et al., 2021*; *Muñoz-Castañeda et al., 2021*; *Sun et al., 2021*; *Chen et al., 2022*) and have validated and extended findings by previous studies using imaging-based tracing techniques.

Despite the transformative power of barcoding-based neuroanatomical techniques, the Sindbis virus-based approaches can only reveal information about the axons (i.e. projection mapping) of labeled neurons with cell bodies at the injection sites, because these vectors do not spread between cells. These approaches thus cannot obtain information about the neurons' synaptic partners. A potentially powerful extension of this approach that could allow mapping of connections, instead of just projections, would be to use rabies virus. Rabies virus naturally spreads between synaptically-connected neurons in the retrograde direction. In particular, the use of deletion-mutant rabies viruses (e.g. 'ΔG' viruses, with their glycoprotein gene G deleted) to perform transsynaptic labeling has become common in neuroscience, as it allows 'monosynaptic tracing' (*Wickersham et al., 2007b*; *Wall et al., 2010*), or the viral labeling of neurons directly presynaptic to a targeted starting population of neurons.

In the standard monosynaptic tracing paradigm, a targeted starting population of neurons is first transduced by 'helper' adeno-associated viruses (AAVs), causing them to express the rabies virus gene G (to complement the G-deleted recombinant rabies) and an avian cell surface protein, TVA. TVA is the receptor for an avian retrovirus (avian sarcoma and leukosis virus subgroup A, or ASLV-A) that is unable to infect mammalian cells (*Young et al., 1993*). Subsequently, a ΔG rabies virus that is packaged with the ASLV-A envelope protein (EnvA) is injected at the same location to selectively infect the TVA-expressing cells. Due to the expression of the rabies viral gene G in trans, the ΔG rabies virus replicates and spreads from these 'source cells' to neurons directly presynaptic to them. In a second approach, ΔG rabies virus that is packaged with its own glycoprotein can be used to retrogradely label neurons with projections to a target area (*Wickersham et al., 2007a*). This approach, in contrast to the monosynaptic tracing paradigm, reveals only the axonal projections of neurons, but not their synaptic connectivity.

RNA barcoding can be combined with both rabies virus-based tracing approaches to drastically improve the throughput at which projections or connectivity can be interrogated at cellular resolution. Retrograde labeling from multiple locations using several preparations of glycoprotein-deleted rabies virus (RVΔG), each carrying different barcodes, would allow multiplexed retrograde tracing, that is the association of neurons with their projections to many different injection sites within single brains (*Figure 1A*). This approach is conceptually similar to that used in a recent study in which barcoded AAV was used to perform multiplexed retrograde labeling (*Zhao et al., 2022*), but rabies virus potentially has different tropism that can broaden the use of barcoded retrograde labeling approaches (*Chatterjee et al., 2018*). Multiplexed retrograde tracing approaches complement anterograde tracing techniques: Because only neurons that are labeled at the injection sites are mapped in anterograde tracing, it is difficult to precisely estimate how the density of projection neurons varies across

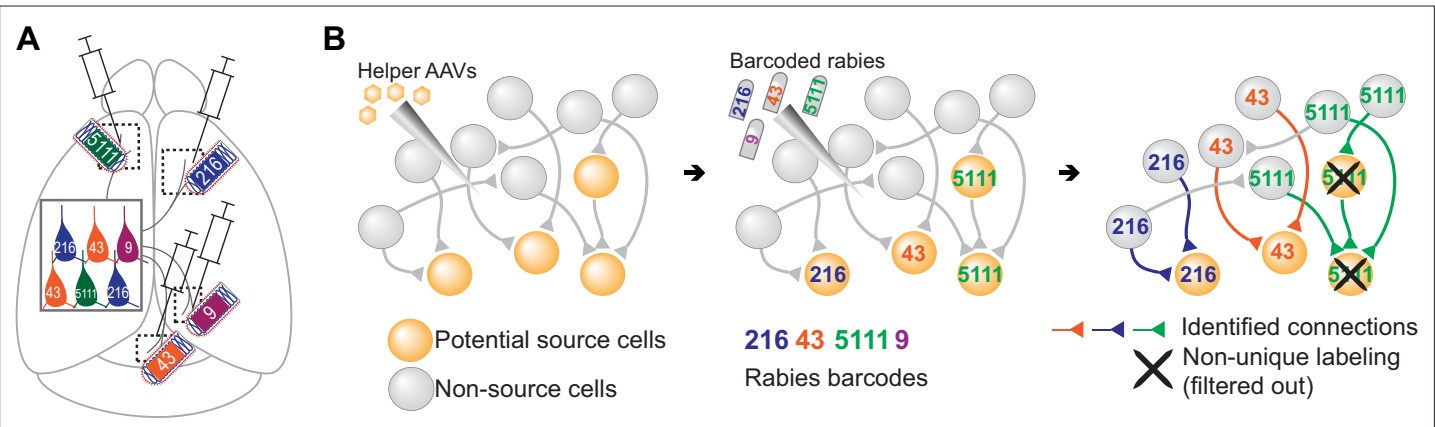

**Figure 1.** Models of multiplexed retrograde labeling and monosynaptic tracing using barcoded rabies virus. (**A**) In multiplexed retrograde labeling, rabies viruses carrying different barcodes are injected into different brain regions, and retrogradely labeled neurons can be distinguished based on the barcodes they carry. Numbers and colors indicate different barcodes injected into each area. (**B**) In multiplexed monosynaptic tracing, potential source cells are first labeled by helper AAV viruses expressing TVA and rabies glycoprotein (yellow cells, *left*). These cells can be infected by barcoded rabies virus (barcodes are indicated by numbers, *middle*). These source cells can then pass the barcodes to presynaptic neurons (numbered gray cells, *right*). Both rabies barcodes and endogenous mRNAs can be read out to infer cell type connectivity. However, if multiple source cells share the same barcode, they may obscure single-cell connectivity mapping and must be filtered out (crossed out cells, *right*).

large brain regions; in contrast, retrograde labeling approaches label neurons with projections to the injection site regardless of which brain region they are in. As anatomical borders between brain areas (e.g. cortical areas) usually correspond to distinct changes in cytoarchitecture (*Brodmann, 1909*; *Vogt and Vogt, 1919*; *Von Bonin, 1947*) and transcriptomically defined neuronal types (*Chen et al., 2023*), retrograde labeling is particularly valuable for understanding how long-range connectivity is associated with cell types and anatomical boundaries.

Alternatively, combining the transsynaptic labeling approach with RNA barcoding could allow sequencing-based readouts of synaptically-connected networks of neurons, providing information about the synaptic partners of neurons, rather than only their patterns of axonal projections. Specifically, simply using the standard monosynaptic tracing system but using a barcoded pool of ΔG rabies virus could allow mapping of synaptic connectivity between many cells at cellular resolution (multiplexed transsynaptic labeling; *Figure 1B*). Although other techniques, such as electron microscopy (*Bae et al., 2021*; *Androvic et al., 2022*; *Zheng et al., 2022*; *Schneider-Mizell et al., 2023*) and multi-patch experiments (*Campagnola et al., 2022*), can also read out synaptic connectivity, these approaches are difficult to scale up to many neurons across large areas of the brain. Thus, achieving barcoded transsynaptic tracing using rabies virus could potentially transform the scale at which synaptic connectivity of transcriptomic types of neurons can be interrogated at cellular resolution.

Using barcoded rabies virus-based approaches to associate connectivity or projections of neurons with their gene expression requires overcoming several technical and conceptual challenges. First, rabies virus alters the gene expression of infected cells (*Patiño et al., 2022*), which may obscure the transcriptomic signatures of neurons. It is unclear how changes in gene expression affect the ability to distinguish fine-grained transcriptomic types and whether cell typing is still possible using spatial transcriptomic approaches, such as in situ sequencing, that interrogate a targeted panel of genes. Second, conventional single-cell sequencing is costly and labor intensive. Thus, sequencing throughput will likely limit the multiplexing advantages associated with barcoding strategies. Third, in a barcoded transsynaptic labeling experiment, synaptic connectivity between individual source cells and their presynaptic partners can only be inferred from networks of barcode-sharing neurons with a single source cell (*Figure 1B*). A barcode, however, may be found in multiple or no source cells, which would prohibit connectivity mapping at the single-neuron level. Finally, accurately assessing the distribution of barcodes in a transsynaptic labeling experiment requires sampling most barcodes and/or source cells in an experiment. This is difficult to achieve using single-cell sequencing approaches that require tissue dissociation, because loss of cells during dissociation is inevitable. These challenges are fundamental barriers for using barcoded transsynaptic labeling to infer synaptic connectivity at cellular resolution regardless of the choice of virus. Solving these problems will not only allow barcoded rabies virus-based connectivity mapping, but also provide a foundation for potential future techniques based on a wide range of transsynaptic viruses, such as herpes simplex virus (*Ugolini et al., 1989*; *Xiong et al., 2022*; *Fischer et al., 2023*), pseudorabies virus (*Martin and Dolivo, 1983*), vesicular stomatitis virus (*Beier et al., 2013*), and yellow fever virus (*Li et al., 2021*). Because in situ sequencing (*Ke et al., 2013*; *Chen et al., 2023*) can read out both endogenous mRNAs and random RNA barcodes (*Chen et al., 2019*; *Sun et al., 2021*) with high throughput and low cost and does not rely on tissue dissociation (*Chen et al., 2019*; *Sun et al., 2021*), it is uniquely suited to overcome the challenges associated with barcoded rabies tracing.

Here we adapt single-cell RNA-seq and an in situ sequencing approach based on Barcoded Anatomy Resolved by Sequencing (BARseq) (*Chen et al., 2019*; *Sun et al., 2021*) to map connectivity of transcriptomic types of neurons using barcoded rabies-based retrograde labeling and transsynaptic labeling. We examine the effect of rabies virus infection on the gene expression and clearly identify transcriptomic identities of rabies-labeled neurons. We then explore conceptually how connectivity can be inferred from rabies barcodes in a trans-synaptic labeling experiment. Finally, we perform scRNA-seq and in situ sequencing on neurons that are transsynaptically labeled by rabies virus to identify pairs of cell types that show preferences to synapse onto the same post-synaptic neurons.

## Results

### Identifying transcriptomic types of retrogradely labeled neurons by barcoded rabies virus

To assess whether we can robustly identify transcriptomically defined neuronal types in a multiplexed retrograde labeling experiment, we performed two-plex retrograde labeling using two libraries of barcoded rabies virus coated with the native rabies glycoprotein (*Figure 2A*; Materials and methods M4). In addition to encoding the red fluorescent protein mCherry (*Shaner et al., 2004*), the two viral libraries contained 20-nt barcode cassettes located in the 3'UTR of the rabies nucleoprotein mRNA, which allowed high-level expression of the barcodes (*Conzelmann, 1998*). In addition, one of the libraries contained a 22-nt exogenous sequence (the 10 x Genomics 'Chromium capture sequence 2', referred to below as CCS) next to the barcode cassette. We sequenced the two barcoded libraries using Illumina next-generation sequencing and identified at least 8552 and 13,211 barcodes, respectively (see Materials and methods M8 and Materials and methods M9; *Figure 2B*). We did not find barcodes that were present in both libraries. Thus, the two libraries could be distinguished both by their barcode sequences and the presence or absence of the CCS.

We injected the two libraries into the dorsal lateral geniculate nucleus (LGd) and anterolateral visual cortex (VISal), respectively, in two animals (*Figure 2C*). After 7 days, we dissected out the primary visual cortex (VISp), dissociated the neurons, and FACS-isolated 48 mCherry-expressing neurons from each animal for scRNA-seq using SMART-seq v4 (see Materials and methods M6). scRNA-seq data from rabies virus-infected cells had comparable quality to non-infected cells (*Figure 2—figure supplement 1A*). Of 96 mCherry-expressing cells sequenced, 75 neurons were of high quality (with >100,000 total reads,>1,000 detected genes, and odds ratios of GC dinucleotides <0.5; see Materials and methods M7 for details).

We then mapped each neuron to reference scRNA-seq data (*Tasic et al., 2018*) by comparing a cell's marker gene expression with the marker gene expression of the reference cell types (*Gouwens et al., 2020*; *Graybuck et al., 2021*) (Materials and methods M7). Briefly, we first selected marker genes that distinguished each cluster in the reference taxonomy tree, and then performed 100 rounds of correlation analysis between a given single cell transcriptome to be mapped and the reference taxonomy tree using 80% of the marker panel selected at random in each round. Cells that were assigned to the same cluster in ≥70 of 100 rounds (mapping confidence >0.7) and with a mapping correlation >0.6 were considered to be mapped to the cluster with high quality. A total of 54 cells were mapped to the reference taxonomy tree with high quality [mapping confidence >0.7 (5 cells removed) and mapping correlation >0.6 (an additional 16 cells removed); *Figure 2D*; *Figure 2—figure supplement 1B*; *Table 1*; Materials and methods M7]; these cells were used for downstream analysis. Most cells that were removed by the mapping correlation thresholds were either 'Microglia Siglech' or 'PVM Mrc1', which tend to have lower gene counts compared to neurons (*Tasic et al., 2018*). Consistent with previous studies (*Prosniak et al., 2001*; *Zhao et al., 2011*; *Huang and Sabatini, 2020*; *Patiño et al., 2022*), some immune response-related genes were up-regulated in rabies virus-infected cells compared to non-infected cells (*Figure 2—figure supplement 2A, B*). We also noticed that some inhibitory cell types showed higher expression of activity-related genes, such as *Baz1*, *Fosl2*, and *Jun* compared to non-infected cells (*Figure 2—figure supplement 3*). Nonetheless, the expression patterns of cell type markers were comparable to the reference scRNA-seq dataset (*Figure 2E and F*). Thus, transcriptomic types of rabies infected neurons can be robustly read out using scRNA-seq.

Rabies transcripts were detected in all FACS-collected cells, including the 54 mapped cells (*Figure 2G*). Consistent with their known expression levels (*Conzelmann, 1998*), transcripts for the rabies nucleoprotein, phosphoprotein, and matrix proteins were more abundant than the transcripts for mCherry (which replaced the rabies glycoprotein) and the large protein. Consistent with the robust detection of rabies transcripts, 50 out of 54 neurons had sequencing reads covering the barcode region of the nucleoprotein transcript (*Figure 2H*; *Table 1*). Among the 50 neurons, 48 neurons carried one unique barcode each, and two cells shared a barcode. Of these 49 barcodes, 18 had the CCS sequence at the 3' flanking region. The barcode sequences were consistent with known barcodes in the corresponding barcode libraries: all 18 CCS-containing barcodes matched known barcodes in the CCS library and all 31 barcodes that did not contain a CCS sequence matched known barcodes in the non-CCS library. In the cortex, L6 corticothalamic (CT) neurons and L5 extra-telencephalic neurons (L5 ET, also known as L5 pyramidal tract/PT neurons) mainly project to the thalamus, but not the

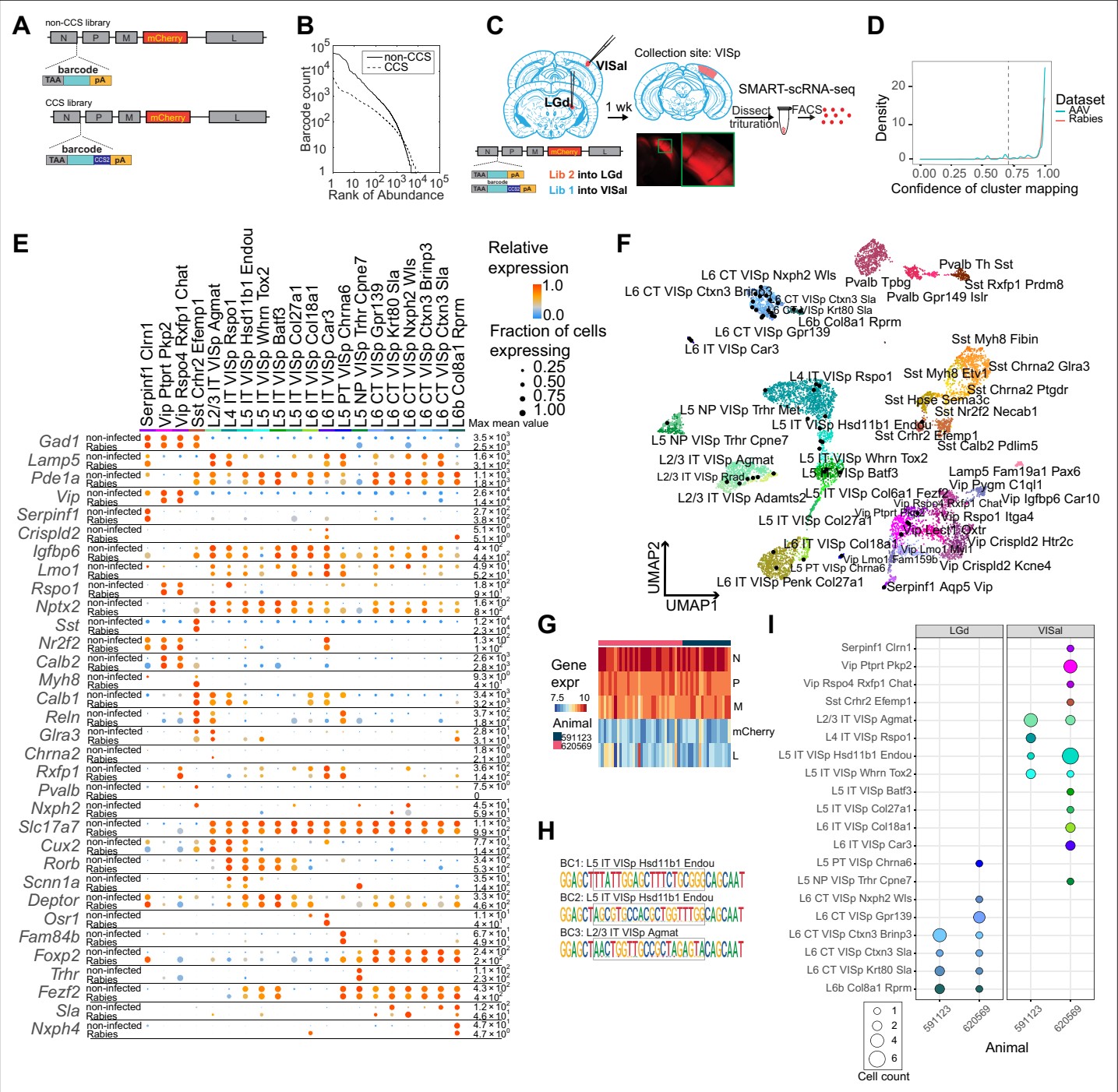

**Figure 2.** High-quality cell typing of rabies virus-barcoded neurons using single-cell RNA-seq. (**A**) Illustration of the design of barcoded rabies virus libraries. (**B**) Barcode distribution in the CCS and non-CCS libraries. The y axis indicates the count of barcodes, which are sorted in descending order. The rank of the barcode is shown on the x axis. (**C**) An outline of the experiments. The two libraries are injected into VISal and LGd. After a week, VISp is dissected, and the mCherry-expressing cells are FACS-enriched and processed for single-cell RNA-seq. (**D**) Histogram showing the cluster mapping confidence of rabies-labeled cells from this study and AAV-labeled cells from *Graybuck et al., 2021*. (**E**) The expression of select marker gene in non-infected cells from *Tasic et al., 2018* and in rabies-infected cells of matching cell types. Dot size indicates the proportion of cells with non-zero marker expression, and colors indicate relative gene expression levels scaled per row. (**F**) UMAP plot of gene expression patterns of cells infected with rabies virus (black) overlaid on non-infected cells from *Tasic et al., 2018*. The non-infected cells are color-coded by cluster identities and cluster names are indicated. (**G**) The expression of rabies-encoded genes in the sequenced cells. Columns indicate cells and rows indicate genes. Colors indicate the log transformed count for each rabies-encoded gene after scaling by the sum of all reads that mapped to viral constructs multiplied by 10,000. The bar on top indicates donor animals. (**H**) Example barcode sequences in three sequenced cells. Letter heights indicate probabilities at each position. Gray boxes

*Figure 2 continued on next page*

*Figure 2 continued*

indicate the barcode region. (**I**) Distribution of cell types of retrogradely labeled cells. Colors indicate cell types and match those in (**F**), and dot size indicates the number of cells. VISp, primary visual cortex; VISal, anterolateral visual cortex; LGd, dorsal lateral geniculate nucleus.

The online version of this article includes the following figure supplement(s) for figure 2:

**Figure supplement 1.** Quality control of scRNA-seq in rabies virus-infected cells.

**Figure supplement 2.** The expression of immune response-related genes in rabies virus-infected cells.

**Figure supplement 3.** The expression of activity-related genes in rabies virus-infected cells and uninfected cells from *Tasic et al., 2018*.

cortex, whereas intra-telencephalic (IT) neurons mainly project to the cortex and the striatum, but not the thalamus (*Harris and Shepherd, 2015*; *Harris et al., 2019*; *Peng et al., 2021*). Consistent with the known connectivity of cortical cell types (*Harris and Shepherd, 2015*; *Tasic et al., 2018*), 14 L6 CT neurons, 3 L6b neurons, and 1 L5 ET neurons projected to the LGd, and not VISal; in contrast, 24 IT neurons and 1 near-projecting (NP) neuron projected to VISal, but not LGd (*Figure 2I*). We also observed a small number of inhibitory neurons labeled by the VISal injection, but not the LGd injection (*Figure 2I*). This labeling was expected, because inhibitory neurons at the border of VISp and VISal could potentially be labeled by the VISal injection and subsequently dissected for single-cell RNA-seq. Based on these results, we estimate that the false positive rate in distinguishing the transcriptomic types of retrogradely labeled neurons is between 0 and 3.1% (see Materials and methods M13). Thus, multiplexed retrograde tracing using barcoded rabies virus recapitulated known projection patterns of cortical neuronal types.

## In situ sequencing identifies transcriptomic types of rabies-barcoded neurons

scRNA-seq provides a detailed view of the transcriptomic landscape of rabies-labeled neurons, but this approach has several limitations. First, a large fraction of neurons is lost during single-cell dissociation prior to sequencing. Because neuronal types differ in their survival rates during dissociation, this procedure introduces biases in the composition of neurons that are sequenced (*Tasic et al., 2016*; *Tasic et al., 2018*). Second, a consequence of tissue dissociation is that the precise locations of neurons are lost, which can obscure potential spatial organization of neuronal connectivity. Finally, the low throughput and high cost of scRNA-seq limit the scale at which the retrogradely labeled neurons can be interrogated. To overcome these limitations, we next used in situ sequencing to resolve both gene expression and rabies barcodes in retrogradely labeled neurons.

Our in situ sequencing approach is based on BARseq, a high-throughput technique that can determine both long-range projections of neurons and their gene expression by in situ sequencing of both endogenous gene expression and mRNA barcodes encoded in the genome of Sindbis virus (*Chen et al., 2019*; *Sun et al., 2021*). We have previously shown that BARseq-style in situ sequencing can distinguish transcriptomic types of cortical neurons with high transcriptomic resolution in non-infected mouse brains (*Chen et al., 2023*). To adapt BARseq to rabies barcodes (*Figure 3A*), we used

**Table 1.** Number of cells in the scRNA-seq-based transsynaptic tracing and retrograde tracing experiments.

|  | Animal ID | N cells | After QC (total reads, genes detected, and GC content) | Mapping confidence > 0.7 | Mapping correlation > 0.6 |
|---|---|---|---|---|---|
| Transsynaptic tracing | 591121 | 94 | 80 | 77 | 75 |
|  | 618308 | 80 | 71 | 69 | 64 |
|  | 618309 | 60 | 46 | 45 | 42 |
|  | 620588 | 61 | 55 | 53 | 51 |
| Retrograde tracing | 591123 | 48 | 34 | 30 | 21 |
|  | 620569 | 48 | 41 | 40 | 33 |
|  | Total | 443 | 371 | 354 | 295 |

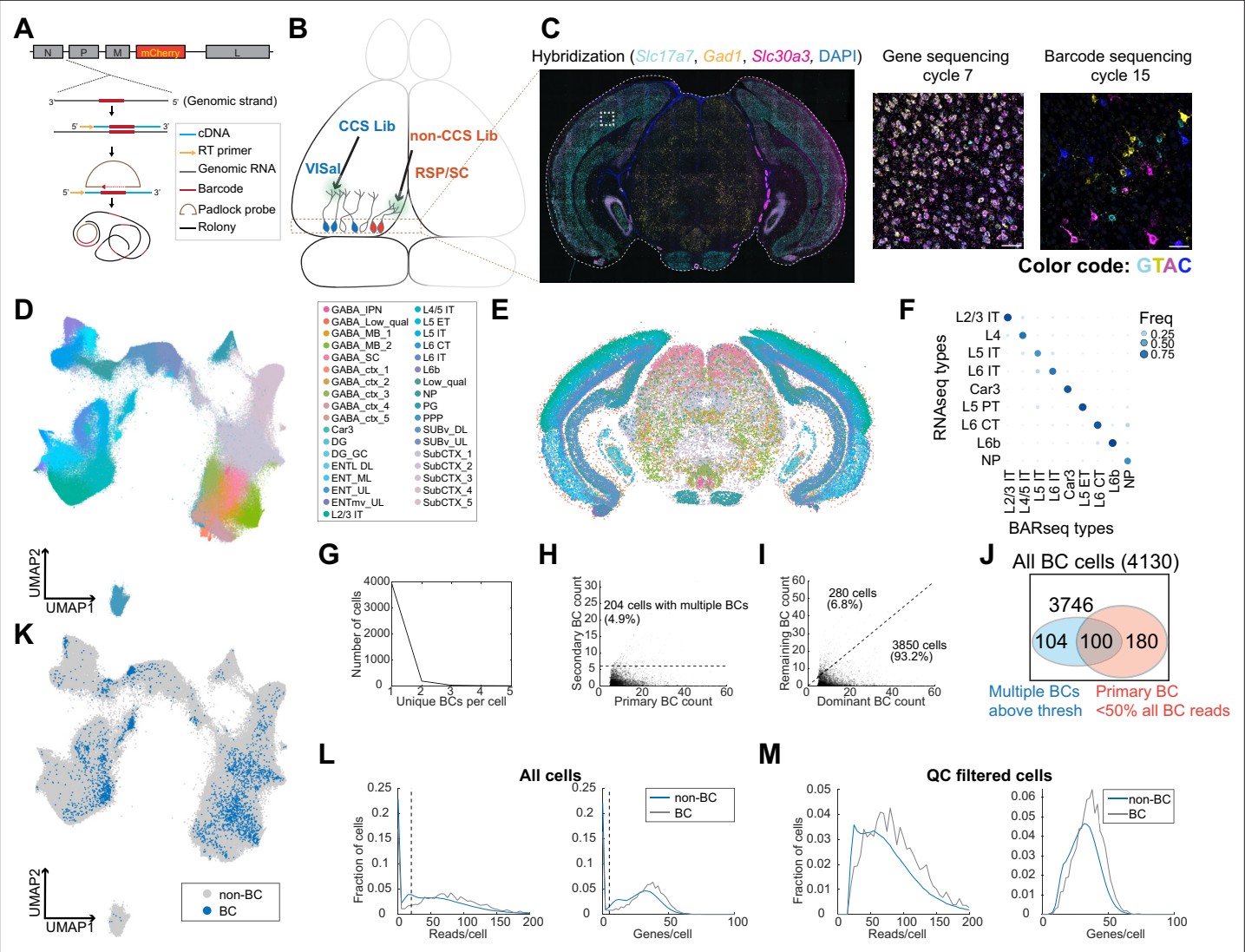

**Figure 3.** In situ sequencing identifies transcriptomic types of neurons infected with barcoded rabies virus. (**A**) Illustration of probe designs and amplification approach for in situ sequencing of rabies barcodes. (**B**) Illustration of the experiments. The two libraries were injected in VISal and RSP/SC, and coronal sections from the boxed area were sequenced. Neurons that project to the two injection sites would be labeled with different sets of barcodes, as indicated by different colors. (**C**) *Left*, example image of a coronal section (outlined by a dashed line) during the hybridization cycle. The final sequencing cycles for genes and barcodes in the boxed area are shown on the *right*. The gene or nucleotide that corresponds to each color is indicated. Scale bars=50 μm. See Supplementary Tabel 3 for a list of genes interrogated. (**D-E**) All sequenced cells shown on a UMAP plot of their gene expression patterns (**D**) or on a representative coronal section (**E**). Colors indicate subclass-level cluster labels as shown in the legend. (**F**) Cluster matching between BARseq subclass-level clusters and subclasses from scRNA-seq (*Tasic et al., 2018*). Dot size and colors indicate the frequency that neurons from a BARseq cluster are assigned to a reference scRNA-seq cluster. (**G**) The number of cells (y axis) with the indicated number of barcodes per cell (x axis). (**H**) The count of the primary barcode (x axis) and the count of the second most abundant barcode (y axis) in each barcoded cell. Cells above the dotted line are considered as having more than one barcode per cell. (**I**) Counts of the most dominant barcode (x axis) and the remaining barcodes (y axis) in each cell. (**J**) Summary of the number of cells with more than one barcodes (blue) and/or cells in which the primary barcodes accounted for less than half of all barcode reads (red). The shapes are not drawn to scale. (**K**) UMAP plot as plotted in (**D**), color coded by whether the cells had barcodes. (**L-M**) Histograms showing the distribution of endogenous mRNA reads per cell and unique gene counts per cell in all cells (**L**) or the QC-filtered cells (**M**). In (**L**), the dotted vertical lines indicate QC thresholds. BC, Barcoded cells; VISal, anterolateral visual cortex; RSP, retrosplenial cortex; SC, superior colliculus.

primers that were complementary to the region that was 3' to the barcodes on the genomic strand to reverse-transcribe the rabies barcode. In addition to the primers for the rabies barcodes, we also used random 20-mers to reverse transcribe the endogenous mRNAs. After reverse transcription in situ, we used padlock probes to target both the flanking regions of the barcodes and 104 marker genes

(Supplementary Tabel 3), which we previously used to distinguish cortical excitatory neuron types (*Chen et al., 2023*). We then gap-filled the barcode-targeting padlock probes, ligated all padlock probes, and performed rolling circle amplification as in standard BARseq experiments (*Sun et al., 2021*). We performed seven rounds of Illumina sequencing in situ to read out 101 cell type marker genes, one round of hybridization to detect three high-level cell type markers (*Slc17a7*, *Gad1*, and *Slc30a3*), and 15 rounds of sequencing in situ to read out the rabies barcodes. We segmented the cells using Cellpose (*Stringer et al., 2021*) and decoded gene rolonies using BarDensr (*Chen et al., 2018*; *Chen et al., 2019*) to obtain single-cell gene expression, as described previously (*Chen et al., 2023*). To basecall barcode rolonies, we picked peaks in the first round of barcode sequencing images using both an intensity threshold and a prominence threshold. We then determined the pixel values in all four sequencing channels across all cycles to readout barcode sequences (*Chen et al., 2018*; *Chen et al., 2019*), and assigned barcode rolonies to segmented cells. All slices were registered to the Allen Common Coordinate Framework v3 (*Wang et al., 2020*) using QuickNII and Visualign (*Puchades et al., 2019*; Materials and methods M12).

We next applied this approach to examine the distribution of neurons with projections to VISal and the retrosplenial cortex (RSP), two cortical areas within the ventral and dorsal stream of the visual pathways, respectively. We used the same two libraries of ΔG rabies virus that were coated with the rabies virus glycoprotein as in the scRNA-seq based experiment. We injected the library that contained CCS into VISal, and the non-CCS-containing library into RSP; the injection into RSP also infected a portion of the superior colliculus (SC; *Figure 3B*). After 7 days, we sacrificed the animals and cryo-sectioned the brains into 20 μm-coronal sections (*Figure 3C*). We sequenced 791,966 segmented cells on eight coronal sections from two animals, and 524,952 cells had sufficient mRNA reads (at least 20 reads per cell and at least 5 genes per cell) to pass quality control (QC). We performed de novo clustering on QC-filtered cells (Materials and methods M12) (*Chen et al., 2023*) and identified 35 subclasses of neurons, including 20 subclasses of excitatory neurons. We further clustered the 20 excitatory subclasses into 116 types (*Figure 3D and E*). We then mapped the subclasses from our dataset to subclasses in reference scRNA-seq data (*Tasic et al., 2018*) using SingleR (*Aran et al., 2019*, *Figure 3F*). Consistent with previous BARseq results (*Chen et al., 2023*), the BARseq subclasses had one-to-one correspondence to subclasses of cortical excitatory neurons found in scRNA-seq data (*Economo et al., 2018*; *Tasic et al., 2018*). Thus, in situ sequencing resolved cortical excitatory neurons at high transcriptomic resolution.

To assign barcodes to cells, we considered a barcode to be associated with a cell if we found at least six molecules of that barcode in a cell (Materials and methods M13). Of 4130 barcoded cells we identified, all but 204 (4.9%) cells contained a single barcode that was above this threshold (>6 counts per cell; *Figure 3G and H*). In most cells, the primary barcode, that is the barcode with the most abundant counts in each cell, accounted for 82% ± 19% (mean ± standard deviation) of barcodes within each cell; in 3850 out of 4130 (93.2%) barcoded cells, the read counts of the primary barcode accounted for at least half of all barcode reads in a cell before thresholding (i.e. including barcodes with six or fewer reads in a cell; *Figure 3I and J*). Consistent with the scRNA-seq experiment described above (*Figure 2F*), barcoded cells co-mingled with non-barcoded cells in a UMAP plot (*Figure 3K*) and had similar read counts and gene counts per cell compared to non-barcoded cells (*Figure 3L and M*), suggesting that labeling by rabies virus did not significantly alter the expression of the genes that we targeted. The barcoded cells included excitatory neurons in the cortex, and inhibitory neurons in the cortex and in the midbrain regions; these inhibitory neurons likely picked up barcodes through their local axons, because the injection sites were either adjacent to or in the same cortical and subcortical areas that were sequenced in situ. Thus, consistent with the results of the scRNA-seq-based approach (*Figure 2*), in situ sequencing resolved the transcriptomic identities of rabies infected cells and determined their rabies barcodes.

## In situ sequencing-based retrograde tracing reveals projections of transcriptomic types of neurons across cortical areas

From all barcoded cells, we identified 1415 unique barcodes and matched them to the two barcoded libraries (containing at least 8552 and 13,211 barcodes, respectively; see Materials and methods M9). Because we sequenced 15 out of 20 bases in the rabies barcodes, we first assessed whether the shorter reads were sufficient to unambiguously assign barcodes to the two libraries. We calculated the

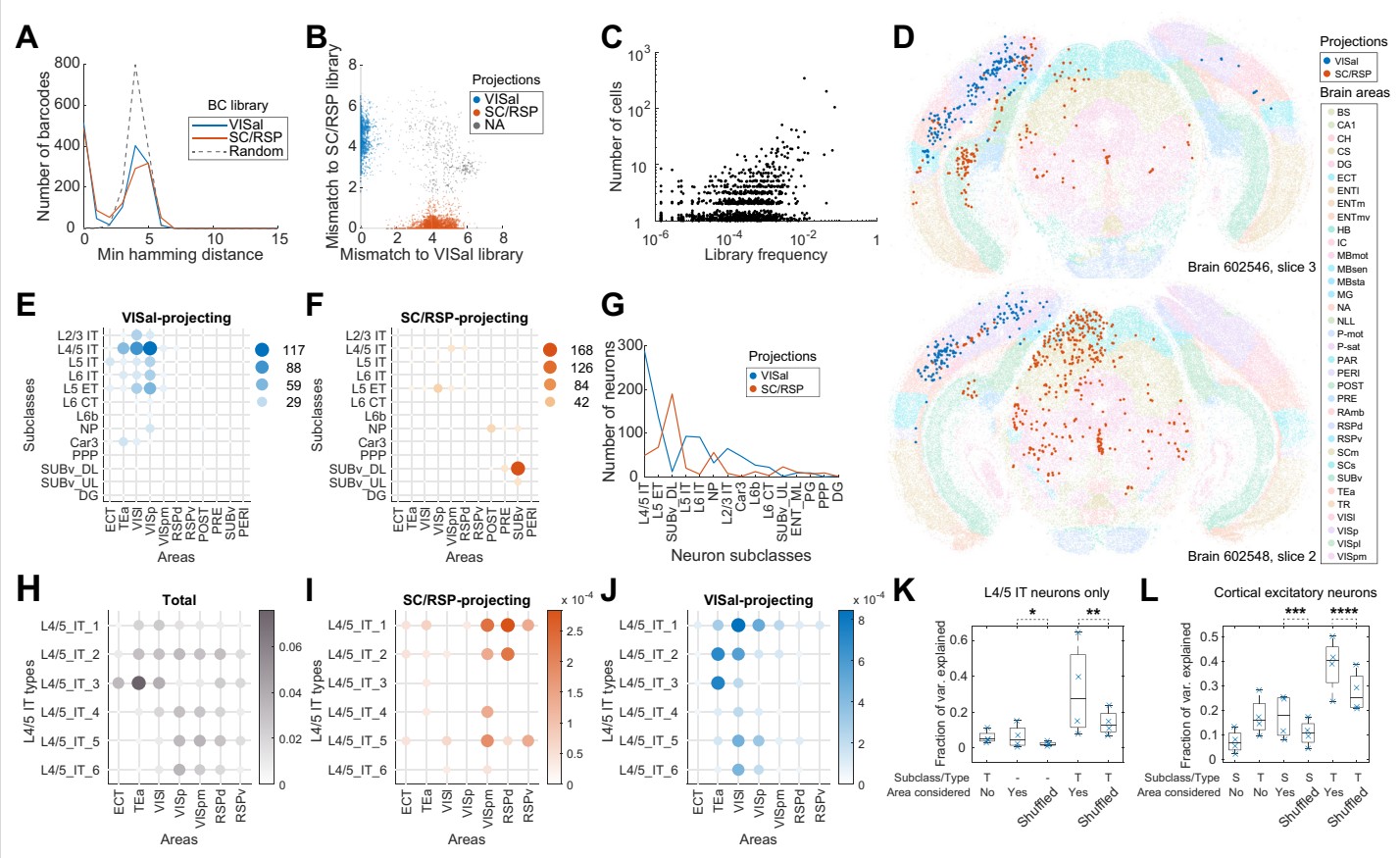

**Figure 4.** Multiplexed retrograde labeling recapitulates known cortical projections. (**A**) Histograms showing the distribution of the minimum Hamming distance between each barcode and all other barcodes for barcodes in the VISal library (blue), the SC/RSP library (red), and random barcodes (dashed). (**B**) The distribution of mismatch between each sequenced barcode and the closest barcode in the CCS library (y axis) or in the non-CCS library (x axis). Colors indicate which library each barcode was mapped to. Gray cells did not match to either library, likely because the libraries were not sequenced to completion. (**C**) The frequencies of barcodes in the libraries (x axis) are plotted against the number of sequenced cells carrying those barcodes (y axis). Jitter is added on the y-axis to help visualize overlapping dots. (**D**) Two representative slices, one from each brain showing all sequenced cells on each slice. Barcoded cells are color-coded by projections and non-barcoded cells are color-coded by brain regions. (**E-F**) The number of neurons from each cortical area and each subclass that projected to either VISal (**E**) or SC/RSP (**F**). Dot size and colors indicate the number of cells. (**G**) The distribution of the number of projecting cells (y axis) for each subclass (x axis). (**H**)-(**J**) The fractions of the indicated L4/5 IT types in each cortical area (**H**) and the fraction of those that project to SC/RSP (**I**) or VISal (**J**). (**K-L**) Fractions of variance in the probability of projections explained by combinations of the compositional profiles of cell types (S: subclass, T: type, -: cell type was not used) with or without considering cortical areas (Area considered) for L4/5 IT neurons (**K**) or for all excitatory neurons (**L**). Boxes indicate median and quartiles, and whiskers indicate range of data. Each point (N = 4) indicate data from different slides, each containing two slices. *p=6 × 10$^{-70}$, **p=6 × 10$^{-58}$, ***p=7 × 10$^{-78}$, ****p=6 × 10$^{-37}$ comparing the means to 100 iterations of shuffled controls using two-tailed t-tests.

minimum Hamming distance of each barcode found in cells to the known barcodes in the two libraries and a third mock library with similar numbers of random barcodes (**Figure 4A**). Many barcodes had zero to one mismatch to the two barcode libraries, but not a library with similar numbers of random barcodes. This peak was distinct from a second peak that was centered at four to five mismatches and included barcodes that were absent from the barcode libraries. No barcode (out of 1415 unique barcodes in this dataset) was within one mismatch to any barcode in the random barcode library. These results indicate that sequencing 15 bases of the barcodes was sufficient to correct mismatches that could have potentially resulted from errors in sequencing and/or mutation in rabies virus. We thus allowed one mismatch when matching barcodes to the two libraries and found 1169 VISal-projecting cells and 2483 SC/RSP-projecting cells, including five cells that projected to both areas (**Figure 4B**). Of the barcodes that matched the two libraries, the number of cells that carried a barcode correlated with the frequency of that barcode in the library (**Figure 4C**; Pearson correlation 0.41). An additional 496 barcodes from 483 cells did not match either barcode library, likely because the two virus

libraries were not sequenced completely to fully recover all barcodes they contained (see Materials and methods M9 for discussions on the unmatched barcodes).

VISal-projecting cells and SC/RSP-projecting neurons were differentially distributed across brain areas and were enriched in different transcriptomic subclasses (*Figure 4D–G*). VISal-projecting neurons were found mostly in VISp and cortical areas *lateral* to VISp, including VISl and TEa (*Figure 4E*). All transcriptomic types of cortical excitatory neurons, except L6 CT and L6b neurons, were labeled, but the fraction of cells labeled were dependent on the area that the neurons were in. For example, VISal-projecting L2/3 IT neurons were found mostly in VISp and VISl, but not in TEa (*Figure 4E*). Because superficial layer neurons are involved in feedforward projections from lower-hierarchy cortical areas to higher hierarchy ones (*Rockland and Pandya, 1979*), this observation is consistent with the hierarchy of cortical areas VISp and VISl have lower hierarchy than VISal, whereas TEa has higher hierarchy; (*Harris et al., 2019*). In contrast, SC/RSP-projecting neurons were found *medial* to VISp (including VISpm and RSPd), in hippocampal areas including the postsubiculum, presubiculum, and the ventral subiculum (*Figure 4F*), and in the midbrain (*Figure 4D*). Within the cortex, projections were dominated by L4/5 IT neurons in VISpm and RSPd, and L5 ET neurons in VISp. Because IT neurons project to the cortex but not the SC (*Harris and Shepherd, 2015*; *Harris et al., 2019*; *Peng et al., 2021*), the L4/5 IT neurons likely projected to RSP, not SC, whereas, the L5 ET neurons likely projected to the SC. Thus, IT neurons that projected to VISal and RSP were separated along the mediolateral axis. This separation is consistent with the ventral and dorsal streams of the visual pathways (*Wang et al., 2012*): whereas VISl, TEa, and ECT belong to the ventral stream, VISpm and RSPd belong to the dorsal stream. These results recapitulated known patterns of projections across areas and transcriptomic subclasses and were consistent with the two pathways in the mouse visual circuit.

Because neurons of different transcriptomic types within a subclass are differentially enriched across cortical areas (*Yao et al., 2021*; *Chen et al., 2023*), we wondered if the differences in projections of the same subclass of neurons across cortical areas can be explained solely by variations in the composition of cell types within subclasses. We first focused on the L4/5 IT neurons, which project to either VISal or RSP from the lateral or medial cortical areas, respectively. Consistent with our previous observations (*Chen et al., 2023*), the transcriptomic types of L4/5 IT neurons were differentially enriched across cortical areas along the mediolateral axis (*Figure 4H*). For example, L4/5_IT_1 and L4/5_IT_3 neurons were enriched in lateral areas, such as VISl, TEa, and ECT, whereas L4/5_IT_4 to L4/5_IT_6 were enriched in VISp, VISpm, and RSPd. The enrichment of cell types across cortical areas, however, cannot account for the differences in projections of L4/5 IT neurons from different cortical areas (*Figure 4I and J*). For example, L4/5_IT_2 in VISpm and RSPd mainly projected to RSP, but not VISal, whereas those in TEa and VISl projected to VISal, not RSP. To quantify how much source areas and cell types contribute to the diversity in projections, we discretized the cortex into 'cubelets'. Each cubelet spanned all cortical layers, was 20 μm thick along the anteroposterior axis (i.e. the thickness of a section), and was about 110 μm in width along the mediolateral axis (Materials and methods M13). In each cubelet, we calculated the fraction of neurons that projected to VISal for each cell type (we did not perform this analysis for RSP-projecting cells because of insufficient sample size). We then used one-way ANOVA to estimate the variance in the fraction of neurons with projections that can be explained by source area labels, cell type labels, or both (Material and methods M13). If projections are determined solely by cell types and the differences in projection probabilities across cortical areas are only due to different compositions of cell types, then the variance in the fraction of neurons with projections explained by cell types and source areas together should be similar to the variance explained by cell types and shuffled source area labels. Within the L4/5 IT subclass, cell type and source area each explained a modest fraction of variance (cell type: 5.9 ± 3.7%; source area: 6.3 ± 6.5%, mean ±std; *Figure 4K*). In contrast, combining cell type and source area explained more variance (31.8 ± 25.7%, mean ±std; *Figure 4K*;) compared to cell types and shuffled source area labels ($p = 6 \times 10^{-58}$ using one-sample two-tailed t-test comparing to 100 iterations of controls with shuffled areas). Similar results were observed across all subclasses of cortical excitatory neurons (*Figure 4L*; $p = 6 \times 10^{-37}$ using one-sample two-tailed t-test comparing types with areas to 100 iterations of types with shuffled areas). In particular, the additional variance explained by areas is similar in combination with types compared to in combination with subclasses (*Figure 4L*). These results indicate that projections of cortical neurons are determined by both the transcriptomic identities of the neurons at the resolution we observed and their anatomical location.

# Interpreting barcoded transsynaptic tracing requires distinguishing barcode-sharing networks

Because we can identify transcriptomic identities of neurons labeled with rabies barcode, similar approaches can also be used in a transsynaptic labeling experiment to interrogate synaptic connectivity. In a barcoded transsynaptic labeling experiment, networks of neurons can share the same barcode because of either connectivity between presynaptic cells and source cells or technical and biological artifacts. Distinguishing different types of barcode-sharing networks is thus crucial for resolving single-cell connectivity. In the following, we first lay out how different types of barcode-sharing networks affect connectivity mapping and how they can be distinguished from each other. We then use in situ sequencing to resolve these networks in a barcoded transsynaptic labeling experiment.

Ideally, each source cell should express a unique barcode, which is also shared with its presynaptic cells. By matching barcodes in the presynaptic cells to those in the source cells, we can then infer connectivity between the presynaptic cells to individual source cells (*single-source networks*; *Figure 5Aa*). In practice, however, multiple source cells may express the same barcode. This barcode sharing could be caused by two viral particles with the same barcodes that independently infected two source cells (*double-labeled networks*; *Figure 5Ab*). For a given barcode, the probability that the same barcode is found in a second source cell because of double labeling scales with the frequency of that barcode in the library and the total number of source cells that are directly labeled in an experiment. Thus, for a given experiment with known number of source cells, barcodes that are more abundant in a library are more likely to result in double-labeled networks. This type of barcode sharing can be minimized by employing sufficiently diverse and uniformly distributed barcodes (*Kebschull et al., 2016*). However, this is typically challenging to achieve with rabies virus, because the recombinant rabies virus production process usually leads to uneven amplification of barcodes, resulting in their highly skewed distribution (*Clark et al., 2021*; *Saunders et al., 2022*). Alternatively, two cells expressing the rabies glycoprotein can be interconnected. In this case, the source cell may pass the barcode to the other cell, thus obscuring the identity of the original source cell (*connected-source networks*; *Figure 5Ac*). Because the probability of connected-source networks scales with both the number of source cells and their local connection probability, limiting the number of source cells or restricting source cells to sparsely connected subpopulations of neurons will reduce this type of network. Finally, it is possible to find barcodes that occur only in cells without glycoprotein expression, but not in corresponding source cells. This can happen when the rabies virus directly infects cells that do not express the glycoprotein (*no-source networks*; *Figure 5Ad*). This may happen because the virus library contains trace amount of non-pseudotyped rabies virus, or because the infected cells express TVA but not the glycoprotein (these are expressed from separate helper AAVs in our system). In this scenario, each barcode should be found in only one or a few cells, because the probability of generating a large no-source network is exponentially smaller: For a barcode with frequency $f$ in the library and total $N$ infection events, the probability of having $M$ independent infection events with that barcode is $CM_N \times f^N \times (1-f)^{M-N}$. As an example, in an experiment with 1000 infection events, the probability for a barcode with $f = 0.001$ to appear in a no-source networks with at least 2 cells is as high as 0.26, but the probability for the same barcode to appear in a no-source network with at least 7 cells is $8 \times 10^{-5}$. Alternatively, source cells may be dead due to cytotoxicity or missed by sequencing (*lost-source networks*; *Figure 5Ae*). If a source cell died, it may spread the barcode to presynaptic neurons and/or to neighboring cells through the release of viral particles during necrosis. In either case, we speculate that these networks are likely to contain more neurons than no-source networks in an experiment with a reasonable number of total infections. Furthermore, the presence of lost-source networks in an experiment could cause some double-labeled networks or connected-source networks to be misidentified as single-source networks because some source cells in these networks may have been lost.

Because only true single-source networks can unambiguously resolve synaptic connectivity between individual source cells and presynaptic cells, mapping connectivity between source cells and presynaptic cells requires an experiment in which all source cells are sequenced and few source cells are lost. In contrast, single-source networks and lost-source networks can all be used to infer synaptic convergence, that is the degree to which two neuronal types synapse onto the same cells, regardless of the identity of the source cell. Because the probability of generating no-source networks and double-labeled networks for a barcode can both be estimated given the total number of barcodes

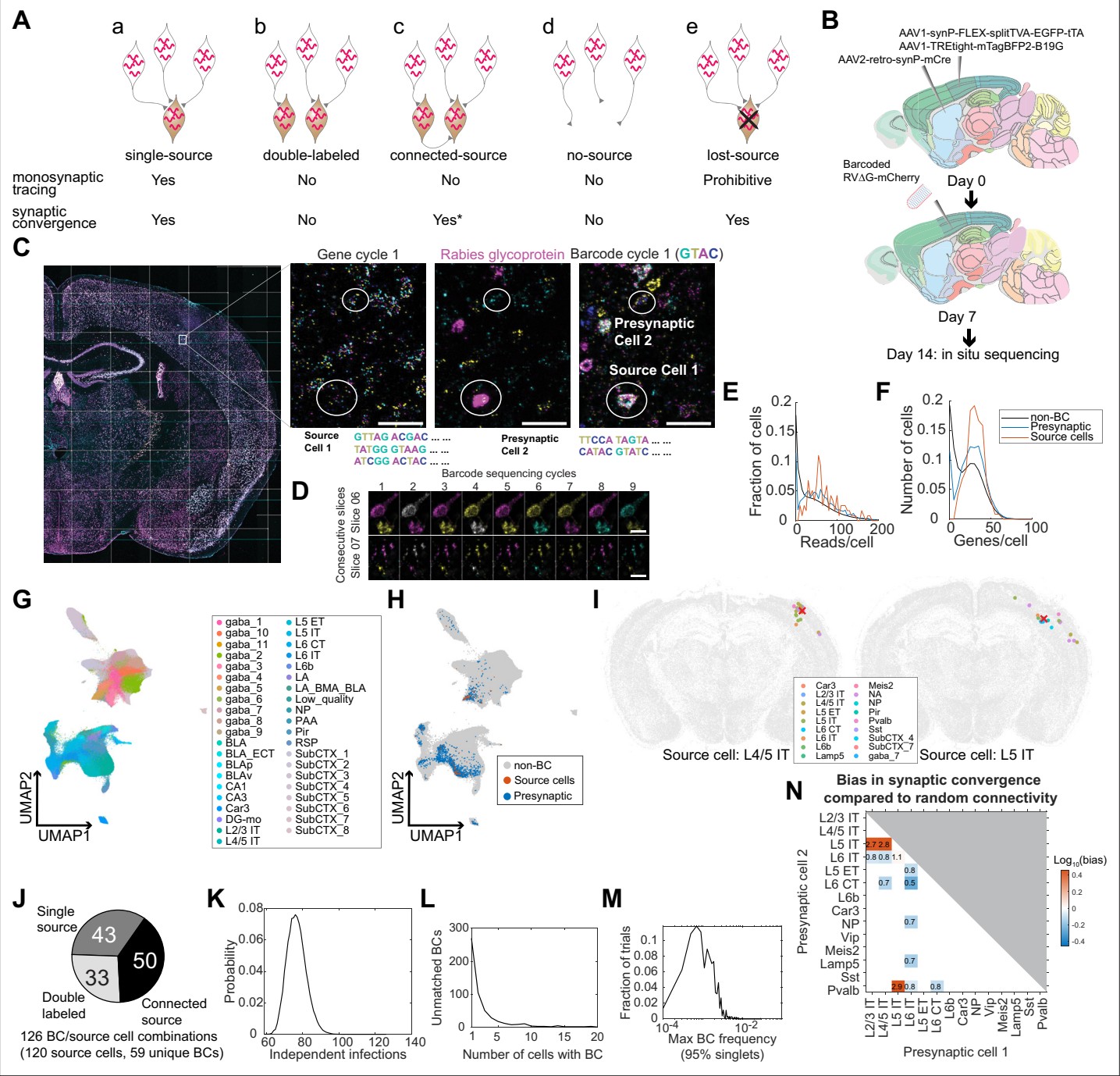

**Figure 5.** Multiplexed transsynaptic labeling by sequencing rabies barcodes in situ. (**A**) Five possible types of barcode-sharing networks in a barcoded transsynaptic tracing experiment using rabies virus. Whether each network is compatible with monosynaptic tracing and/or mapping synaptic convergence is indicated below. *see text for considerations regarding connected-source networks. (**B**) Summary of the transsynaptic labeling experiment using barcoded rabies virus and in situ sequencing. (**C**) The image of a representative coronal section during sequencing. Images of the first gene sequencing cycle, the hybridization cycle, and the first barcode sequencing cycle of the boxed area are shown on the right. Scale bars=50 μm. (**D**) First nine barcode sequencing images of example neurons that were bisected onto two adjacent sections. Scale bars = 10 μm. (**E-F**) The distribution of endogenous mRNA reads per cell (**E**) and unique gene counts per cell (**F**) in non-barcoded cells (gray), barcoded cells (blue), and source cells (red). Dashed lines indicate quality control thresholds for gene expression. (**G-H**) UMAP plots of the gene expression patterns of all barcoded cells color-coded by the cluster label at the subclass level (**G**) or by whether the cell is a potential source cell or presynaptic cell (**H**). (**I**) Locations and cell types of two source cells (red cross) and presynaptic cells (dots) that shared the same barcodes. Colors of dots indicate transcriptomic types of presynaptic neurons. Transcriptomic types of source cells are indicated below each plot. All other cells from the coronal sections that the source cells were on were

*Figure 5 continued on next page*

plotted in gray. (**J**) Estimated numbers of barcode/source cell combinations that belonged to each of the three networks with source cells. (**K**) The probability (y axis) of the number of independent infection events (x axis) to generate the same number of barcodes found in the source cells in the experiment. (**L**) Histogram showing the number of cells that shared each barcode that was not found in a source cell. (**M**) Histogram showing the distribution of the maximum barcode frequency in the virus library that ensures single infection for 95% of barcodes across 10,000 simulations. (**N**) The ratios between the observed number of converging outputs and the expected number from random connectivity between cortical subclasses of neurons. Colors correspond to log10 of ratios, and the ratios are indicated in the plot. Only values with false positive rate (FPR) < 0.05 are shown (see Materials and methods M14). As seen from the blue squares associated with L6 IT cells, these neurons were less likely to synapse onto the same post-synaptic neurons with other neuronal types (in particular L6 CT neurons, with only 50% of converging connections compared to those expected from random connectivity).

The online version of this article includes the following figure supplement(s) for figure 5:

**Figure supplement 1.** scRNA-seq is insufficient to resolve connectivity among transsynaptically labeled neurons using barcoded rabies virus.

**Figure supplement 2.** Barcoded transsynaptic labeling resolved by in situ sequencing.

**Figure supplement 3.** Presynaptic cells and source cells in all single-source networks.

seen in an experiment and the frequency of barcodes in the library, these networks can be identified and filtered out if most barcodes are sequenced in an experiment. Connected-source networks may in principle obscure synaptic convergence if a source cell that is transsynaptically labeled further pass the barcode to its presynaptic cells. However, because these tertiary infections (presynaptic cells of the transsynaptically labeled source cell) occur after the secondary infections (presynaptic cells of the original source cell), the number of barcoded cells that are presynaptic to the transsynaptically labeled source cell is likely smaller compared to those that are presynaptic to the original source cell. We thus speculate that connected-source networks are dominated by presynaptic cells of the original source cells and can also be used in the analysis of synaptic convergence. Therefore, inferring synaptic convergence neither requires sequencing all source cells nor is constrained by the presence of lost source cells.

## In situ sequencing of barcoded rabies virus reveals cell type preferences in synaptic convergence

Having established how barcode-sharing networks can be analyzed in theory, we next applied this approach to a barcoded trans-synaptic rabies tracing experiment. We first tried using SMART-seq v4 scRNA-seq to read out the connectivity and transcriptomic identity of transsynaptically labeled neurons from four animals (*Figure 2—figure supplement 1*). However, because scRNA-seq can only interrogate a fraction of all barcoded cells, we were unable to confidently infer connectivity from the rabies barcodes (see Appendix 1 for details; *Figure 5—figure supplement 1*). To sequence more barcoded cells, we then investigated whether in situ sequencing could be used in a multiplexed trans-synaptic labeling experiment (*Figure 5B*). We injected AAV2-retro-syn-mCre (*Jin et al., 2023a*) into the dorsal striatum, and two helper viruses, AAV1-syn-FLEX-splitTVA-EGFP-tTA and AAV1-TREtight-mTagBFP2-B19G (*Liu et al., 2017*), into the somatosensory cortex. This combination allows expression of TVA and the rabies glycoprotein in corticostriatal neurons (*Jin et al., 2023a*). After seven days, we injected a barcoded EnvA-pseudotyped ΔG rabies virus library into the somatosensory cortex. This library contained the same barcodes as the CCS-containing library used for retrograde labeling, with at least 13,211 barcodes. After another seven days, we collected 20 μm coronal sections and sequenced both endogenous genes and the rabies barcodes in 32 consecutive sections in situ (*Figure 5C*). These sections, which spanned 640 μm along the AP axis, largely covered the whole injection site. Because in situ sequencing did not require dissociation and was able to capture even both halves of barcoded cells that were split between two sections (*Figure 5D*), sequencing consecutive sections that spanned the whole injection site should allow us to interrogate most source cells. We interrogated the same gene panel as in the retrograde labeling experiment, except that we substituted probes for *Slc30a3* with probes for the rabies glycoprotein gene. Barcode rolonies were basecalled individually within cells, which allowed us to read out multiple barcodes from the same cell (*Figure 5C*).

We sequenced 3,107,645 cells, including 2914 barcoded cells (summarized in *Table 2*). Barcoded cells were defined as cells with at least eight counts of the same barcode, with the barcode being sufficiently complex (linguistic sequence complexity >$10^{-0.9}$; see Materials and methods M14 for definition). These quality control thresholds were determined by manually proofreading a subset of barcoded cells:

**Table 2.** The numbers of barcoded cells that belonged to each type of network in the in situ sequencing-based trans-synaptic tracing experiment.

| | Total | Single-source | Double-labeled | Connected-source | No-source | Lost-source |
|---|---|---|---|---|---|---|
| Source cells and barcodes | 120 cells, 59 barcodes, 126 cell-barcode pairs | 42 cells, 43 barcodes, 43 cell-barcode pairs | Est. 33 cell-barcode pairs | Est. 50 cell-barcode pairs | 0 | 0 |
| presynaptic cells | 2590 cells (=381 + 979–6+677 + 566–7) | 381 cells | 979 cells (6 cells also contained a single-source barcode) | | 677 cells (7 cells also had a lost-source barcode) | 566 cells (7 cells also had a no-source barcode) |
| Barcodes in presynaptic cells | 535 barcodes (=31 + 16+427 + 61) | 31 barcodes | 16 barcodes | | 427 | 61 |
| Filtered out cells | 204 cells (with one G transcript and/or low-quality source cells) | NA | NA | NA | NA | NA |

two proofreaders examined barcoded cells with different barcode counts or barcode complexity and determined whether each cell was indeed barcoded; based on the manual inspections, we then determined the best threshold that allowed us to distinguish cells from background fluorescence and from barcodes in passing axons/dendrites (*Figure 5—figure supplement 2A*). We next defined potential source cells and potential presynaptic cells based on rabies glycoprotein expression. In conventional trans-synaptic tracing experiments, potential source cells are usually determined by the expression of fluorescent proteins. Because the in situ sequencing approach we used did not preserve fluorescent labeling, we used the copy numbers of the rabies glycoprotein transcript to distinguish source cells from presynaptic cells. We estimated a threshold for the glycoprotein transcript, so that the transcriptomic types of cells with glycoprotein reads above this threshold were consistent with corticostriatal projection neurons (see Appendix 2 and *Figure 5—figure supplement 2* for details). We found that cells expressing at least 12 counts of the glycoprotein transcript likely correspond to BFP + cells in a conventional tracing experiment. However, we also observed many cells with lower numbers of glycoprotein transcripts (see Discussion about the potential sources of this low-level expression). Although these cells were unlikely to be considered source cells in a conventional tracing experiment, the glycoprotein expressed in them could potentially allow barcodes to be passed onto presynaptic neurons. Thus, excluding them from potential source cells may cause double-labeled networks and connected-source networks to be mistaken as single-source networks (see Appendix 2 and *Figure 5—figure supplement 2* for a detailed analysis). To minimize potential false-positive identification of synaptic connections, we used a conservative threshold of two transcripts per cell to define potential source cells. Of the 2914 barcoded cells, 138 cells had at least two counts of the rabies glycoprotein and were considered potential source cells; 2590 barcoded cells had no glycoprotein expression and were considered potential presynaptic cells; the remaining 186 barcoded cells had a single read of glycoprotein and were removed from further analyses. Consistent with the retrograde labeling experiment, the barcoded cells had similar numbers of endogenous gene reads as non-barcoded cells (*Figure 5E and F*) and clustered together with non-barcoded cells (*Figure 5G and H*).

Because distinguishing different types of barcode-sharing networks requires sequencing most source cells, we first estimated what fraction of the source cells that were present in the tissue was sequenced. The number of cells that shared a barcode varied widely from 1 cell/barcode to 359 cells/barcode. Since previous studies showed that each source cell can label dozens to hundreds of presynaptic neurons (*Liu et al., 2013*; *Wertz et al., 2015*) when infected at relatively high viral titer (as we have done in this experiment), we reasoned that barcodes that were found only in a small number of cells may indicate either no-source networks or lost-source networks due to dead source cells. We thus focused on 23 barcodes that were present in at least 12 cells per barcode, and calculated the fraction of barcodes that were also found in source cells with read counts above a certain threshold. We varied this threshold to generate a range of estimates that likely over-estimated and under-estimated the number of source cells at the extreme ends, respectively, and found that 80–93% barcodes were found in source cells (see Materials and methods M14 for details; *Figure 5—figure supplement 2B*). We further filtered the 138 potential source cells to remove those that had wrong segmentations or included barcodes from nearby cells based on manual examination of the sequencing images, resulting in 120 high-quality source cells with 59 unique barcodes.

We then estimated the number of barcodes that belonged to the three types of networks with source cells (single-source networks, double-labeled networks, and connected-source networks; *Figure 5Aa-c*). Six source cells contained two barcodes each and the remaining 114 source cells contained a single barcode, resulting in 120 source cells and 126 unique combinations of source cell/barcodes. Forty-three barcodes in 42 source cells were found only in a single source cell each (one of these source cells contained two barcodes); of these 43 barcodes, 31 barcodes in 30 source cells were shared with 381 putative presynaptic cells (12±14 cells per barcode after rounding, mean ± std; *Figure 5I*; *Figure 5—figure supplement 3*). The remaining 16 (59 – 43) barcodes labeled multiple source cells each (81 source cells in total, including 3 source cells that contained both shared barcodes and unique barcodes).

Barcodes shared across source cells could be caused by either connected source cells (*Figure 5A and C*) or double labeling (*Figure 5A and B*). To estimate the contribution of these two sources, we leveraged the fact that the probability of double labeling scales with the frequency of a barcode in a library, whereas the probability of connected source cells does not. Because source cell/barcode

combinations, rather than barcode counts or source cell counts alone, represented potential independent infection events or transsynaptic labeling between source cells, we used the number of source cell/barcode combinations to estimate the number of barcodes of either type of network. Briefly, we calculated the posterior probability of $N$ transduction events to generate 59 barcodes in the source cells, given the barcode frequency in the library (Materials and methods M14). We found that the most likely number of transduction events was 76 (95% confidence interval 67–87; *Figure 5J and K*), suggesting that about 33 (76 – 43) barcode/source cell combinations can be explained by multiple source cells being infected with the same barcodes through independent infection events. Since there were a total of 126 unique combinations of source cells and barcodes, the remaining 50 (126 - 76) source-barcode pairs were likely from source cells that were connected to other source cells (*Figure 5J*). This high number of connected source cells is expected, because many cortical neuronal types are highly connected (*Campagnola et al., 2022*). Thus, our analyses suggest that barcode sharing among source cells was dominated by connected-source networks.

We next estimated the number of barcode-sharing networks without source cells (no-source networks or lost-source networks; *Figure 5Ad*, e). To distinguish these networks, we took advantage of the fact that no-source networks, which are created by multiple infections involving the same barcode, are exponentially rarer as the size of the network increases; in contrast, lost-source networks, which are created by spreading from missing source cells, are not expected to follow this relationship. In total, 490 barcodes were found in presynaptic cells but not in source cells. Indeed, the distribution of the number of cells containing each barcode that were absent in source cells revealed two peaks, one large peak around 1 cell per barcode and a smaller peak around 8 cells per barcode (*Figure 5L*). Of these barcodes, 429 (88%) were found in fewer than five cells, suggesting that barcodes that were not found in source cells were dominated by direct infection of cells that did not express the glycoprotein (no-source networks). To estimate a lower bound of the number of source cells that died, we considered all networks with at least five cells and no source cell as lost-source networks. This conservative estimate resulted in 61 dead source cells, compared to 120 observed source cells and 76 estimated independent infection events. These results are consistent with previous observations using single cell-initiated monosynaptic tracing that around half of source cells died by seven days after infection (*Wertz et al., 2015*). Because many source cells died in this experiment, some barcodes that were originally shared across multiple source cells may appear as single-source networks. This ambiguity obscures mapping synaptic connectivity between the source cells and presynaptic cells, but has limited effect on inferring synaptic convergence, that is which neuronal types are more likely to synapse onto the same post-synaptic cell, regardless of the identity of the post-synaptic cell.

We next filtered our data with the goal of estimating synaptic convergence. As discussed above, single-source networks (*Figure 5Aa*), lost-source networks (*Figure 5Ae*), and connected-source networks can all be used to estimate synaptic convergence, because presynaptic neurons that shared the same barcodes all synapsed onto the same cells (although the identity of the post-synaptic cells are unknown in lost-source networks). No-source networks (*Figure 5Ad*) and double-labeled networks (*Figure 5Ab*), in contrast, contained many neurons that were not related by connectivity and would obscure synaptic convergence. To minimize the impact of these two networks, we identified and removed both types of networks by limiting network size and barcode frequency, respectively. Because cells in no-source networks were generated by independent infection events and generally contained few cells, we first filtered out networks that were too small (<5 cells; Materials and methods M14). Because double-labeled networks likely occurred to barcodes that were abundant, we applied a maximum barcode frequency threshold ($1.3\times10^{-3}$ based on the library sequencing). This threshold should ensure that about 95% of the barcodes could only have transduced a single source cell from the initial infection by EnvA-enveloped virus (see Materials and methods M14; *Figure 5M*).

For all potential presynaptic neurons that shared a barcode, we counted each pair of presynaptic neurons as a pair of 'converging connections'. We aggregated all pairs of converging connections across all filtered barcodes for cortical subclasses of presynaptic neurons, and calculated the bias in synaptic convergence as the ratio between the observed number of converging connections compared to the expected numbers of connections based on random connectivity (*Figure 5N*; see Materials and methods M14). That is, if neurons of subclass A and those of subclass B are more likely to synapse onto the same post-synaptic cell, then these subclasses have a convergence bias greater than 1, and vice versa. We found that IT neurons generally synapse onto similar cells (the convergence

bias is 2.75 between L2/3 IT and L5 IT, 2.81 between L4/5 IT and L5 IT, and 1.06 between L5 IT and L6 IT, false positive rate <0.05 by shuffled control). However, IT neurons in layer 6 (L6 IT) were less likely to synapse onto the same postsynaptic cells as other cortical neurons, including IT neurons in the superficial layers (L2/3 IT and L4/5 IT, convergence bias 0.83 and 0.80, respectively), the corticofugal projection neurons (L5 ET and L6 CT, convergence bias 0.78 and 0.53, respectively), NP neurons (convergence bias 0.72), and two subclasses of inhibitory neurons (Lamp5 and Pvalb, convergence bias 0.70 and 0.84, respectively). The differences in the connectivity of L6 IT neurons are reminiscent of their characteristic areal and laminar patterns of long-range projections (*Tasic et al., 2018*; *Chen et al., 2019*) compared to other IT neurons. We speculate that the distinct L6 IT connectivity may reflect specialized roles in cortical processing.

## Discussion

Here, we combined scRNA-seq and in situ sequencing with barcoded rabies virus to relate connectivity and transcriptomic identities of neurons in both multiplexed retrograde labeling and transsynaptic labeling. As a proof of principle, we applied these approaches to interrogate projections and synaptic connectivity of cortical neuronal types. Our experiments recapitulated the diverging dorsal and ventral streams in the visual cortex and revealed differences in projections of neurons across cortical areas and transcriptomic types. Furthermore, we laid out the requirements for achieving and assessing barcoded transsynaptic labeling and revealed converging and diverging synaptic connectivity among cortical neuronal types. Our study provides a proof of principle for applying barcoded rabies virus-based neuroanatomy in the mouse brain in vivo and lays the foundation for achieving multiplexed monosynaptic tracing using barcoded rabies virus.

### In situ sequencing of barcoded rabies virus achieves multiplexed retrograde labeling

In barcoding-based multiplexed retrograde tracing, different barcodes are used to distinguish projections to different injection sites. Unlike conventional fluorescence-based multiplexed retrograde tracing techniques, a barcoding-based approach is not limited by the number of distinguishable colors of fluorescence. Because the maximal barcode diversity that can be theoretically achieved increases exponentially with the length of the barcode (4 N in which *N* is the length of the barcode), possible barcode diversity is virtually unlimited. Thus, in a barcoded retrograde tracing experiment, the number of targets that can be investigated by retrograde labeling in a single experiment is likely limited by the number of injections. In our experiments, we only performed retrograde labeling from two sites as a proof of principle, but this approach can be applied to retrograde tracing from dozens of sites to reveal multiple projections from individual cells in future experiments.

Barcoded retrograde and anterograde projection mapping approaches obtain complementary information. Barcoded anterograde labeling approaches, such as MAPseq and BARseq, can barcode many neuronal somata in a local region in a single injection, and thus can reveal the diversity of projections of densely labeled neurons within a region. In contrast, retrograde labeling approaches can reveal the distribution of neurons with the same projections over large brain regions. In this study, we interrogated the distribution of neurons across seven adjacent cortical areas and found that both source areas and the transcriptomic identities of cortical neurons contribute to their projection patterns. To perform the equivalent experiment using an anterograde tracing approach would require multiple injections at precise locations to achieve uniform labeling across a large region of the cortex. The sensitivity of the two approaches is also constrained by different factors: the sensitivity of MAPseq/BARseq relies on detecting individual barcode molecules in the axons; in contrast, because rabies genome is highly amplified in each infected cell, the sensitivity of barcoded rabies-based retrograde labeling is not limited by barcode detection but determined by viral transduction. Any future improvement in rabies virus-based retrograde tracing in general would also benefit barcoded approaches. Therefore, the combination of barcoding-based retrograde and anterograde projection mapping techniques provides a versatile set of tools for understanding the organization of long-range circuits.

## In situ sequencing resolves barcode-sharing networks in transsynaptic labeling

Transsynaptic labeling using barcoded rabies virus has the potential to map synaptic connectivity of individual neurons with unparalleled throughput and to resolve the transcriptomic identities of connected neurons at high resolution. In a barcoded transsynaptic labeling experiment, accurately inferring synaptic relationships among neurons requires distinguishing different types of barcode-sharing networks (*Figure 5A*) and filtering out networks that can obscure connectivity analysis. Accurately identifying the network that each neuron belongs to requires sequencing all source cells. This requirement is difficult to achieve using single-cell RNA-seq-based approaches (*Clark et al., 2021*; *Saunders et al., 2022*). Because conventional single-cell approaches require tissue dissociation, some source cells are inevitably lost. In situ sequencing, in contrast, bypasses the need for tissue dissociation and has the potential to sequence all source cells. Accidental loss of sections during cryosectioning can be further minimized using a tape-transfer system (*Pinskiy et al., 2015*), which is also compatible with BARseq-style in situ sequencing (*Chen et al., 2019*). In situ sequencing thus provides a path to accurately resolve synaptic connectivity in a barcoded rabies-based transsynaptic labeling experiment.

In our experiment, we found that many source cells likely died in the experiment, which prevented us from unambiguously determining the connectivity between source cells and presynaptic neurons. Nonetheless, we could infer which neuronal types were more likely to synapse together onto the same source cell, regardless of the identity of the source cell. This analysis goes beyond previous barcoded rabies-based approaches (*Clark et al., 2021*; *Saunders et al., 2022*), and represents a first proof-of-principle of inferring the statistics of synaptic connectivity of neuronal types using a barcoding-based approach. Further optimizing the technique to unambiguously determine connectivity between source cells and presynaptic neurons will not only generate insights into the connectivity of neuronal types, but also allow comparison to connectivity inferred from electron microscopy data (*Bae et al., 2021*) and patch-seq data (*Campagnola et al., 2022*) to validate the specificity of rabies monosynaptic tracing.

## Optimizing viral strategies to achieve highly multiplexed mapping of synaptic connectivity

Our analyses based on the in situ sequencing data identified several factors that can be optimized in future experiments to achieve barcoded monosynaptic tracing at single-cell resolution. Most importantly, reducing source cell death would allow more precise matching between pre-synaptic neurons and source cells. Losing source cells not only obscures the various types of barcode-sharing networks but may also produce neurons that share the same barcode not through synaptic connectivity, but through necrosis of the source cell. Cytotoxicity in source cells can be ameliorated by titrating concentrations of AAV and rabies virus (*Lavin et al., 2020*), optimizing the incubation period after injecting rabies virus, using alternative strains of rabies virus with less cytotoxicity (*Reardon et al., 2016*), and/or using engineered glycoprotein (*Kim et al., 2016*). Because barcode sequencing in situ was sufficiently sensitive, our approach can also be adapted to new generations of deletion mutant vectors with reduced viral replication (*Chatterjee et al., 2018*; *Jin et al., 2022*).

In addition to minimizing source cell death, other factors can be optimized to map synaptic connectivity more efficiently. First, we found many rabies-infected neurons with barcodes that were absent in source cells. Although these small no-source networks do not need to be eliminated to map synaptic connectivity, they nonetheless reduce effective barcode diversity and contribute to noise in the data. We speculate that these rabies-infected cells resulted from direct infection by rabies virus in neurons that had a low-level expression of TVA in the absence of Cre. It could be possible to minimize direct infections in future experiments by using mutant TVA that reduces background infection (TVA66T) (*Miyamichi et al., 2013*) and by optimizing the titer of rabies virus and AAV helper viruses (*Lavin et al., 2020*). Furthermore, parallel control experiments, in which AAV helper viruses expressing either the Cre or the glycoprotein is omitted, can be used to establish ground truth for the prevalence of independent infections and no-source networks.

Second, reducing the number of source cells that share barcodes can improve the number of networks that can be used to infer synaptic connectivity. Because the number of connected source cells increases with both the probability of connections and the total number of cells that express

the glycoprotein, connected source cells can be minimized by titrating the number of glycoprotein-expressing neurons and/or by labeling specific subpopulations of neurons that were sparsely connected. Improving the diversity and/or the distribution of barcodes in the rabies library would also reduce barcode sharing across source cells.

Finally, trans-synaptic transmission efficiency can be improved in future experiments. In our experiment, many potential source cells with low numbers of glycoprotein transcripts had only small numbers of corresponding presynaptic cells, suggesting that this low-level expression only allowed transmission of the rabies virus with low efficiency. We speculate that this broad but low-level expression of rabies glycoprotein may have been driven by the TREtight promoter in the absence of Cre-dependent tTA expression (*Loew et al., 2010*). Thus, placing the glycoprotein under the direct control of Cre could reduce its broad but low-level expression and constrain potential source cells to a sparser population. These modifications, combined with control experiments that assess the extent of connected source cells, double labeling, and dead source cells experimentally, could help achieve efficient barcoded monosynaptic tracing in future experiments.

### Building a comprehensive barcoded connectomics toolbox based on in situ sequencing

Barcoding-based monosynaptic tracing has unique advantages compared to other connectivity mapping techniques. Paired recordings combined with patch-seq can determine the functional connectivity of a small number of neurons and their transcriptomic types (*Campagnola et al., 2022*), but it is restricted to mapping connectivity over a short distance, since both connection probability and the probability of preserving those connections in slice culture drops considerably as the patched neurons become more distant. CRACM (channelrhodopsin-assisted circuit mapping) (*Petreanu et al., 2007*) can map long-range synaptic connectivity, but it can only map inputs from a population of genetically labeled neurons and cannot resolve differences within this population. Electron microscopy (*Bae et al., 2021*; *Shapson-Coe et al., 2021*) remains the gold standard for identifying synapses, but it is challenging to associate connectivity with the transcriptomic identities of neurons and to interrogate long-range connectivity across brain regions. In comparison, barcoding-based monosynaptic tracing is compatible with interrogation of transcriptomic identities of neurons. Furthermore, because the connectivity is read out by sequencing instead of tracing, the error in connectivity mapping does not scale with the distance between the connected neurons. These characteristics are not restricted to rabies virus-based monosynaptic tracing, but also apply to potential future techniques based on other trans-synaptic viruses (*Martin and Dolivo, 1983*; *Ugolini et al., 1989*; *Beier et al., 2013*; *Li et al., 2021*; *Xiong et al., 2022*; *Fischer et al., 2023*) Thus, barcoding-based monosynaptic tracing is ideally suited for mapping the long-range synaptic connectivity of neuronal types and complements existing synaptic connectivity mapping tools.

In situ sequencing is ideally suited for barcoding-based neuroanatomical techniques compared to conventional dissociation based single-cell or single-nuclear RNA-seq techniques. In situ sequencing not only has the ability to capture most source cells, but is also faster and cheaper in mapping cell types than most dissociation based techniques on a per-cell basis (*Chen et al., 2023*). For example, in situ sequencing performed in the transsynaptic labeling experiment, which included dense sequencing of 3.1 million cells in total across 32 consecutive coronal sections, cost less than $4,000 in reagents. Furthermore, in situ sequencing captures the mapped neurons at high spatial resolution, which can reveal spatial organization of connectivity. Such space-based rules in connectivity are common in vertebrate brains (e.g. retinotopy, cortical column) and may exist independent of transcriptomic type-based differences in connectivity (*Chen et al., 2022*); specific spatial patterns of connectivity loss are common in many neurodegenerative diseases. By combining in situ sequencing with barcoded rabies tracing, we expand in situ sequencing-based barcoded neuroanatomical approaches to include anterograde projection mapping (BARseq), multiplexed retrograde tracing, and multiplexed transsynaptic tracing. This diverse set of in situ sequencing-based tools enables flexible strategies for mapping neuronal connectivity across scales.

## Materials and methods

**Key resources table**

| Reagent type (species) or resource | Designation | Source or reference | Identifiers | Additional information |
|---|---|---|---|---|
| Strain, strain background (*Mus musculus*) | C57BL/6 J (See *Supplementary file 1* for details) | Jackson Laboratory | 000664 | |
| Recombinant DNA reagent | pRVdG-4mCherry (plasmid) | *Weible et al., 2010* | Addgene_52488 | |
| Recombinant DNA reagent | RabV CVS-N2c(deltaG)-mCherry (plasmid) | *Reardon et al., 2016* | Addgene_73464 | |
| Recombinant DNA reagent | pCAG-B19N (plasmid) | *Chatterjee et al., 2018* | Addgene_59924 | |
| Recombinant DNA reagent | pCAG-B19P (plasmid) | *Chatterjee et al., 2018* | Addgene_59925 | |
| Recombinant DNA reagent | pCAG-B19G (plasmid) | *Chatterjee et al., 2018* | Addgene_59921 | |
| Recombinant DNA reagent | pCAG-B19L (plasmid) | *Chatterjee et al., 2018* | Addgene_59922 | |
| Recombinant DNA reagent | pCAG-T7pol (plasmid) | *Chatterjee et al., 2018* | Addgene_59926 | |
| Recombinant DNA reagent | pCAG-N2cN (plasmid) | This paper | Addgene_100801 | Used in rabies virus rescue (see Materials and methods M5) |
| Recombinant DNA reagent | pCAG N2cP (plasmid) | This paper | Addgene_100808 | Used in rabies virus rescue (see Materials and methods M5) |
| Recombinant DNA reagent | pCAG-N2cG (plasmid) | This paper | Addgene_100811 | Used in rabies virus rescue (see Materials and methods M5) |
| Recombinant DNA reagent | pCAG-N2cL (plasmid) | This paper | Addgene_100812 | Used in rabies virus rescue (see Materials and methods M5) |
| Sequence-based reagent | See *Supplementary file 3* for details | Integrated DNA Technologies | NA | |
| Commercial assay or kit | MiSeq Reagent Nano Kit v2 (300-cycles) | Illumina | MS-103–1001 | |
| Commercial assay or kit | RevertAid Reverse Transcriptase | Thermo Fisher | EP0442 | |
| Commercial assay or kit | RiboLock RNase Inhibitor | Thermo Fisher | EO0384 | |
| Commercial assay or kit | Phusion High-Fidelity DNA Polymerase | Thermo Fisher | F530L | |
| Commercial assay or kit | Ampligase Thermostable DNA Ligase | Biosearch Technologies | A0110K | |
| Commercial assay or kit | RNase H | Qiagen | Y9220L | |
| Commercial assay or kit | Phi29 dna polymerase | Thermo Fisher | EP0094 | |
| Chemical compound, drug | Iodoacetamide, No-Weigh | Thermo Fisher | A39271 | |
| Chemical compound, drug | Bis-PEG9-NHS ester | BroadPharm | BP-21504 | |
| Software, algorithm | MATLAB | Mathworks | RRID: SCR_001622 | |
| Software, algorithm | R | R Project for Statistical Computing | RRID: SCR_001905 | |
| Software, algorithm | Custom scripts for data analysis/processing | This paper | | See M15. Data and Code availability for links |
| Other | AAV2-retro-syn-mCre (virus) | *Jin et al., 2023b* | Addgene_178515 (genome plasmid) | Helper virus for barcoded rabies virus (see *Figure 5B*) |
| Other | AAV1-syn-FLEX-splitTVA-EGFP-tTA (virus) | *Liu et al., 2017* | Addgene_100798 (genome plasmid) | Helper virus for barcoded rabies virus (see *Figure 5B*) |

| Reagent type (species) or resource | Designation | Source or reference | Identifiers | Additional information |
|---|---|---|---|---|
| Other | AAV1-TREtight-mTagBFP2-B19G (virus) | *Liu et al., 2017* | Addgene_100799 (genome plasmid) | Helper virus for barcoded rabies virus (see *Figure 5B*) |
| Other | AAV1-TREtight-mTagBFP2-N2cG (virus) | This paper | Addgene_192838 (genome plasmid) | Helper virus for CVS-N2c strain of rabies virus (see *Figure 5—figure supplement 1*) |
| Other | N2cΔG-4mCherry_CCS2_20nt_HM(EnvA) (virus) | This paper | | Barcoded rabies virus library used for monosynaptic tracing with scRNA-seq (see Materials and methods M5) |
| Other | RVΔG-4mCherry_CCS2_20nt_HM(EnvA) (virus) | This paper | | Barcoded rabies virus library used for monosynaptic tracing with BARseq (see Materials and methods M5) |
| Other | RVΔG-4mCherry_CCS2_20nt_HM(B19G) (virus) | This paper | | Barcoded rabies virus library used for retrograde labeling (see Materials and methods M5) |
| Other | RVΔG-4mCherry_20-mer barcode(B19G) (virus) | This paper | | Barcoded rabies virus library used for retrograde labeling (see Materials and methods M5) |

## M1. Animals and surgery

Animal handling and surgery were conducted according to protocols approved by the Institutional Animal Care and Use Committee (IACUC) of the Allen Institute for Brain Science (protocol number 2106 and 2201) and protocol approved by the MIT Committee on Animal Care (protocol number 2303000498). Animals were housed 3–5 per cage and were on a 12/12 light/dark cycle in an environmentally controlled room (humidity 40%, temperature 21 °C). A full list of animals used are provided in *Supplementary file 1*.

For retrograde tracing, barcoded G-deleted rabies viruses enveloped in native rabies glycoprotein were stereotaxically injected into target brain areas of mice using coordinates obtained from the Paxinos adult mouse brain atlas (*Paxinos and Franklin, 2019*). For retrograde transsynaptic tracing, AAV helper viruses were stereotaxically injected into VISp for the scRNA-seq experiments and SSp/dorsal striatum for the in situ sequencing experiments, followed by the injection of EnvA-pseudotyped barcoded G-deleted rabies virus into the same area two weeks later. Details for all injections are provided in *Supplementary file 1*. Brains were dissected 1 week (7+/−1 days) after rabies virus injection.

## M2. Viruses and constructs

Viruses used are provided in *Supplementary file 2*. For plasmids and vectors used for making barcoded rabies virus, see below (**Constructing barcoded plasmids for rabies virus**).

## M3. In situ sequencing oligos

RT primers, sequencing primers, and padlock probes for both endogenous genes and rabies transcripts are designed as previously described (*Sun et al., 2021*; *Chen et al., 2023*). A list of oligos used in in situ sequencing are provided in *Supplementary file 3*.

## M4. Constructing barcoded plasmids for rabies virus

In order to maximize the copy number of barcode transcripts in rabies virus infected neurons, we inserted a barcode sequence consisting of 20 random nucleotides in the 3' untranslated region of the nucleoprotein gene in pRVΔG-4mCherry (Addgene #52488) (SAD B19 strain) (*Weible et al., 2010*) and RabV CVS-N2c(deltaG)-mCherry (Addgene #73464) (CVS-N2c strain) (*Reardon et al., 2016*) using NEBuilder HiFi DNA Assembly for constructing the barcoded rabies plasmid libraries. The 10 x Genomics 'Chromium Capture Sequence 2' (GCTCACCTATTAGCGGCTAAGG) was included following the barcode in one version of the libraries of each strain.

Cloning steps were as follows:

## Step 1: Construction of pRVΔG-4mCherry library template

In order to reduce contamination of the library by the original plasmids, a jGCaMP7s fragment flanked by PacI and PmeI was inserted in the 3' UTR of the N gene in RabV CVS-N2c(deltaG)-mCherry and pRVΔG-4mCherry.

   Here, we used two approaches to minimize contamination: (1) (2) after long-range PCR with Q5 High-Fidelity 2 X Master Mix, the PCR amplicons were run on a low melting agarose gel for 2 hr in order to separate PCR amplicons from pRVΔG-4mCherry template plasmid in case double digestion from (1) was not sufficient.

## Step 2: Whole RV plasmid PCR

To generate large amounts of linearized rabies vector for barcoded plasmids in the next step, we performed long-range PCR using pRVΔG-4mCherry library template from step 1, linearized with PacI and PmeI, as a DNA template. We then ran the PCR mix on a 0.7% low melting point agarose gel (16520–100, Invitrogen) and the target bands at 14,438 bp were cut and purified with NucleoSpin Gel and PCR Clean-Up kit (740609.250, Takara). The target amplicons were re-concentrated to >300 ng/μL with GlycoBlue Coprecipitant (AM9515, Invitrogen). The long-range PCR primers are listed below:

   N_Barcode_160 bp_rp_57 (26-mer): ATTGGAACTGACTGAGACATATCTCC
   N_Barcode_NheI_fp_58 (28-mer): AGCAATCTACGGATTGTGTATATCCATC

   The long-range PCR was performed as described below:
   Q5 Hot Start High-Fidelity 2 X Master Mix (M0494S, NEB): 12.5 μL; N_Barcode_160 bp_rp_57 (10 uM): 1.25 μL; N_Barcode_NheI_fp_58 (10 uM): 1.25 μL; pRVΔG-4mCherry template with PacI and PmeI double digest (without purification, 8 ng/μL): 1 μL; Nuclease-free water (B1500L, NEB): 9 μL.
   Cycle conditions: (1) 98 °C, 30 s; (2) 98 °C, 5 seconds; (3) 67 °C, 10 s; (4) 72 °C, 5 min; (5) go to step 2–4, 30 times; (6) 72 °C, 2 min.

## Step 3: Barcoding RV vector using NEBuilder HiFi DNA assembly method

A 160 bp Ultramer DNA Oligonucleotides with 20 random nucleotide barcodes sequence (see below) was generated by IDT (Integrated DNA Technologies, Inc, USA). They were inserted into the linearized vector of pRVΔG-4mCherry from step 2 described above using HiFi DNA Assembly (NEB, USA). The HiFi reaction was mixed as described below:

   Re-concentrated pRVΔG-4mCherry PCR amplicons (>300 ng/μL): 1 μL; 160 bp Ultramer DNA Oligonucleotides (0.2 uM): 0.9 μL; Nuclease-free water: 8.1 μL; NEBuilder HiFi DNA Assembly Master Mix (E2621L, NEB): 10 μL.

   This reaction mix was incubated at 50 °C for 1 hr. After the incubation, we concentrated the mix to >130 ng/μL with GlycoBlue Coprecipitant. After electroporation transformation with Endura Electrocompetent Cells (60242–1, Biosearch Technologies, USA), the cells were plated into Nunc Square BioAssay Dishes (Catalog number: 240835, Thermo Fisher Scientific, USA). After growing for 14 hr at 37 °C, the bacterial colonies were scraped using bacti cell spreaders (60828–688, VWR, USA) for plasmid library purification with NucleoBond Xtra Midi EF kit (740420.10, Takara).

   The sequence of the 160 bp Ultramer DNA Oligonucleotides is (note that the Ns were ordered as hand-mixed random bases): GAGATATGTCTCAGTCAGTTCCAATCATCAAGCCCGTCCA AACTCATTCGCCGAGTTTCTAAACAAGACATATTCGAGTGACTCATAAGAAGTTGAATAA CAAAATGCCGGAGCTNNNNNNNNNNNNNNNNNNNNAGCAATCTACGGATTGTGTATATCC Production of the CVS-N2c strain library proceeded similarly but using the following PCR primers:
   N2c whole rabies vector PCR primers:
   N2c_Barcode_fp_59 (26-mer): CCTTTCAAACCATCCCAAATATGAGC
   N2c_Barcode_rp_59 (32-mer): AGTCATTCGAATACGTCTTGTTTAAAAATTCG
   170 bp Ultramer DNA Oligonucleotides for N2c library:
   TTAAACAAGACGTATTCGAATGACTCATAAGGAGTTGATTGACAGGGTGCCAGA     NNNNN NNNNNNNNNNNNNNNNAATCTATAGATTGTATATATCCATCGCTCACCTATTAGCGGCTAA GGATCATGAAAAAAACTAACACTCCTCCTTTCAAACCATCCCAAATATGAG

## M5. Rabies virus production

Barcoded rabies viruses were produced and titered largely as described previously (*Wickersham et al., 2010*; *Wickersham and Sullivan, 2015*), using the barcoded vector genome plasmid libraries described above. For SAD B19 viruses, HEK-293T cells were transfected with the respective genome library along with helper plasmids pCAG-B19N (Addgene cat. # 59924), pCAG-B19P (Addgene cat. # 59925), pCAG-B19G (Addgene cat. # 59921), pCAG-B19L (Addgene cat. # 59922), and pCAG-T7pol (Addgene cat. # 59926), were used for rescue transfections, using Lipofectamine 2000 (Thermo Fisher). For CVS-N2c virus, Neuro2a cells stably expressing the CVS-N2c glycoprotein (N2A-N2cG_02 cells) were transfected with the barcoded N2c library along with helper plasmids pCAG-N2cN (Addgene cat. # 100801), pCAG N2cP (Addgene cat. # 100808), pCAG-N2cG (Addgene cat. # 100811), pCAG-N2cL (Addgene cat. # 100812), and pCAG-T7pol. Details for individual preparations are as follows:

### RVΔG-4mCherry_20-mer barcode(B19G) (used for retrograde labeling)

Supernatants were collected beginning 3 days after transfection and continuing for a total of five days, with supernatants replaced with 13 ml fresh medium until the last collection; each supernatant was filtered and refrigerated until all supernatants were titered together on HEK-293T cells as described (*Wickersham et al., 2010*). BHK cells stably expressing SAD B19 glycoprotein were infected with rescue supernatant at a multiplicity of infection (MOI) of 0.1. 24 hr after infection, medium was replaced with 14 ml fresh medium. Supernatants were collected beginning 24 hr later and continuing every 24 hr for a total of three days, clarified by low-speed centrifugation and filtered as described previously (*Wickersham et al., 2010*) and replaced with 14 ml fresh medium until the final collection. Following titering, 25 ml of supernatant III was ultracentrifuged through 25% sucrose as described (*Wickersham et al., 2010*) and resuspended overnight in 20 µl DPBS, aliquoted and frozen, with a final titer of 4.21e11 iu/mL.

### RVΔG-4mCherry_CCS2_20nt_HM(B19G) (used for retrograde labeling)

Production of this virus was similar to that of the above but using the CCS2-containing SAD B19 plasmid library. The final titer of the concentrated virus was 5.97e11 iu/mL.

### RVΔG-4mCherry_CCS2_20nt_HM(EnvA) (used for monosynaptic tracing with BARseq)

Supernatant I of the passage of RVΔG-4mCherry_CCS2_20nt_HM(B19G) described above was used to infect BHK cell stably expressing a fusion protein of the EnvA envelope protein with the cytoplasmic domain of the SAD B19 glycoprotein at an MOI of 2, as described. 24 hr after infection, cells were washed twice with DPBS, and the medium was replaced with 13 ml fresh medium per plate. 24 hr later, cells were again washed once with DPBS, and medium was again replaced. 24 hr later, supernatants were collected, replaced with fresh medium, clarified and filtered, incubated with 30 U/mL Benzonase for 30 min at 37 °C, then ultracentrifuged through sucrose. 24 hr later, a second supernatant was collected and processed the same way. Both concentrated supernatants were then pooled, mixed, and aliquoted as 5 µl aliquots, then stored at –80 °C before being titered on TVA-expressing cells as described previously (*Wickersham et al., 2010*). The final titer of the concentrated stock was 7.16e10 iu/mL.

### N2cdG-4mCherry_CCS2_20nt_HM(EnvA) (used for monosynaptic tracing with scRNA-seq)

Following transfection, medium was changed regularly with 25 ml fresh medium until 10 days post-transfection, when supernatant collection was begun. Supernatants were collected each Monday, Wednesday, and Friday, replaced with 25 ml fresh medium except for the final collection, clarified, filtered, and stored in 4 °C, for a total of nine collections. Supernatants were titered on HEK 293T cells as described previously. N2A-EnvA_cytG cells (*Reardon et al., 2016*) were infected with rescue supernatants IV and V at an MOI of 2. 24 hr after infection, supernatants were aspirated, cells were washed twice with DPBS and given 12 ml fresh medium per plate. 24 hr later, cells were again washed once with DPBS and given 12 ml fresh medium per plate. 24 hr later, supernatants were collected, replaced with fresh medium, clarified and filtered, incubated with 30 U/mL Benzonase for 30 min at 37 °C, then ultracentrifuged through sucrose. 24 hr later, a second supernatant was collected and

processed the same way. The two concentrated supernatants were pooled, aliquoted, and stored at –80 °C. The final titer of the concentrated stock was 2.82e10 iu/mL.

> Primers for N2c virus_Miseq:
> For RT-PCR:
> Adaptor_UMI_N-16_N2c:
> ACACTCTTTCCCTACACGACGCTCTTCCGATCTNNNNNNNNNNNNNNNNNNNNNNATTGACAGGGTGCCAG
> Primers for MiSeq sample preparation: i5-anchor_CTAGCGCT_fp_56:
> AATGATACGGCGACCACCGAGATCTACACCTAGCGCTACACTCTTTCCCTACACGAC i7_TGACAAGC_rp_N2c:
> CAAGCAGAAGACGGCATACGAGATGCTTGTCAGTGACTGGAGTTCAGACGTGTGCTCTTCCGATCTGGTGAGCGATGGATATATACAATCTATAGATT
> Sequencing primers for Miseq:
> Read1 Sequencing primer:
> ACACTCTTTCCCTACACGACGCTCTTCCGATCT
> Read2_sequencing primer_N2c:
> ACGTGTGCTCTTCCGATCTGGTGAGCGATGGATATATACAATCTATAGATT

## M6. Tissue processing and scRNA-seq

scRNA-seq was performed using the SMART-Seq v4 kit (Takara Cat#634894) as described previously (*Tasic et al., 2018*). Mice were anaesthetized with isoflurane and perfused with cold carbogen-bubbled artificial cerebrospinal fluid (ACSF). The brain was dissected and sliced into 250 μm coronal sections on a compresstome (Precisionary). Regions of interest were micro-dissected, followed by enzymatic digestion, trituration into single cell suspension, and FACS analysis. For retrograde tracing, 48–52 mCherry-positive cells from each mouse were sorted into eight-well strips containing SMART-Seq lysis buffer with RNase inhibitor (0.17 U/μL; Takara Cat#ST0764). For retrograde transsynaptic tracing, 48 mCherry-positive cells from each mouse were first sorted into eight-well strips, followed by the sorting of mCherry-/mTagBFP2-double positive cells. Sorted cells were immediately frozen on dry ice for storage at −80 °C. Reverse transcription, cDNA amplification, and library construction were conducted as described previously (*Tasic et al., 2018*). Full documentation for the scRNA-seq procedure is available in the 'Documentation' section of the Allen Institute data portal at http://celltypes.brain-map.org/.

## M7. RNA-seq data processing

Reads were aligned to GRCm38 (mm10) using STAR v2.5.3 in twopassMode as described previously (*Tasic et al., 2018*), and all non-genome-mapped reads were aligned to sequences encoded by the rabies and AAV helper viruses. PCR duplicates were masked and removed using STAR option 'bamRemoveDuplicates'. Only uniquely aligned reads were used for gene quantification. Exonic read counts were quantified using the GenomicRanges package for R. To determine the corresponding cell type for each scRNA-seq dataset, we utilized the scrattch.hicat package for R (*Tasic et al., 2018*). The mapping method was based on comparing a cell's marker gene expression with the marker gene expression of the reference cell types. Selected marker genes that distinguished each cluster were used in a bootstrapped centroid classifier, which performed 100 rounds of correlation using 80% of the marker panel selected at random in each round. Cells meeting any of the following criteria were removed:<100,000 total reads, <1000 detected genes (with CPM >0), CG dinucleotide odds ratio >0.5, mapping confidence <0.7, and mapping correlation <0.6. One animal in which 31 out of the 40 mapped cells were microglia was excluded from analysis.

To extract CCS sequences, non-genome-mapped reads were aligned to the CCS sequence encoded in the rabies genomic sequence, and to extract barcode sequences, non-genome-mapped reads were aligned to the region in the rabies genomic sequence with extra six nucleotides flanking the barcode sequence. genomic sequence. For each sample, nucleotide frequencies of the reads over the specified regions were calculated using the alphabetFrequencyFromBam function of the GenomicAlignments package for R. Position weight matrix (PWM) was constructed from the nucleotide frequency matrix using the makePWM function of the SeqLogo R package, and consensus sequences were derived from PWM objects to represent CCS and barcode sequences.

## M8. Sequencing of barcoded rabies virus libraries

For the SAD B19 viruses, rabies genomes were exacted by NucleoSpin RNA Virus (740956.50, Takara), RT-PCR was performed with AccuScript PfuUltra II RT-PCR Kit (600184, Agilent) and a customized RT-primer including unique molecular identifier (UMI) sequence, named Adaptor_UMI_N15. The UMI-based counting was used for analyzing rabies barcode diversity.

> Adaptor_UMI_N15: ACACTCTTTCCCTACACGACGCTCTTCCGATCTNNNNNNNNNNNN NNNNNNNNNACAAAATGCCGGAGC

The redundant Adaptor_UMI_N15 primers in cDNA sample were removed using NucleoSpin Gel and PCR Clean-Up kit with the ratio NTI solution 1:4. The 217 bp of MiSeq sample was prepared by PCR with a high-fidelity polymerase (Invitrogen Platinum SuperFi II Green PCR Master Mix, 12-369-010, USA). The PCR was performed as described below: Invitrogen Platinum SuperFi II Green PCR Master Mix: 12.5 μL; i5-anchor_CTAGCGCT_fp_56 (10 μM): 1.25 μL; i7_TTGGACTT_rp (10 μM): 1.25 μL; cDNA of rabies genome: 1 μL (>100 ng/μL); Nuclease-free water (NEB): 9 μL. The cycle conditions were: (1) 98 °C, 30 s; (2) 98 °C, 5 s; (3) 60 °C, 10 s; (4) 72 °C, 3 s; (5) go to step 2–4, 16 times; (6) 72 °C, 5 min.

> The pair of primers used are listed below: i5-anchor_CTAGCGCT_fp_56: AATGATACGGCG ACCACCGAGATCTACACCTAGCGCTACACTCTTTCCCTACACGAC
> i7_TTGGACTT_rp: CAAGCAGAAGACGGCATACGAGATAAGTCCAAGTGACTGGAGTTCAGA CGTGTGCTCTTCCGATCTGGATATACACAATCCGTAGATTGC

After running the product on a 3% low melting agarose gel and the target bands (217 bp) were extracted, the samples were cleaned with Unclasping Gel and PCR Clean-Up kit. The purified products were sequenced on MiSeq in MIT BioMicro Center with sequencing primers below:

> Read1 Sequencing primer: ACACTCTTTCCCTACACGACGCTCTTCCGATCT
> Read2 Sequencing primer: TGTGCTCTTCCGATCTGGATATACACAATCCGTAGATTGCT

N2c viruses were sequenced following the same procedures, using the following primers:

> For RT-PCR: Adaptor_UMI_N-16_N2c: ACACTCTTTCCCTACACGACGCTCTTCCGATCTNNN NNNNNNNNNNNNNNNNATTGACAGGGTGCCAG
> Primers for MiSeq sample preparation: i5-anchor_CTAGCGCT_fp_56: AATGATACGGCGACCA CCGAGATCTACACCTAGCGCTACACTCTTTCCCTACACGAC
> i7_TGACAAGC_rp_N2c: CAAGCAGAAGACGGCATACGAGATGCTTGTCAGTGACTGGAGTT CAGACGTGTGCTCTTCCGATCTGGTGAGCGATGGATATATACAATCTATAGATT
> Sequencing primers for Miseq: Read1 Sequencing primer: ACACTCTTTCCCTACACGACGCTC TTCCGATCT
> Read2_sequencing primer_N2c: ACGTGTGCTCTTCCGATCTGGTGAGCGATGGATATATACA ATCTATAGATT

## M9. Sequencing depth of rabies virus libraries

We found 13,212 and 8,553 barcodes, respectively, from the RVΔG-4mCherry_20-mer barcode(B19G) and RVΔG-4mCherry_CCS2_20nt_HM(EnvA) libraries. The RVΔG-4mCherry_CCS2_20nt_HM(B19G) library was used to prepare the RVΔG-4mCherry_CCS2_20nt_HM(EnvA) library, so we did not separately sequence the barcodes in the B19G coated library.

In the retrograde labeling experiment, 12% of barcodes (496 out of 4,130 barcoded cells) did not match barcodes that were seen in the two libraries. These unmatched barcodes were likely caused by the shallow depth at which these two libraries were sequenced. For example, in the RVΔG-4mCherry_CCS2_20nt_HM(EnvA) library, 99.9% of reads corresponded to unique UMIs (200,021 reads corresponding to 199,815 UMIs). This high fraction of reads with unique UMIs indicates that the sequencing was very shallow and were unlikely to reveal all barcodes. As a point of reference, in typical MAPseq and/or BARseq experiments using barcoded Sindbis virus, each unique UMI that corresponds to a barcode molecule usually has at least 5–10 reads (*Kebschull et al., 2016*; *Chen et al., 2019*). Because of this shallow sequencing depth, a large fraction of UMIs were likely missed during sequencing. Because 37% (5914 out of 13,212) and 46% (3964 out of 8553) barcodes found in the two libraries

had a single UMI recovered, many barcodes with similar frequencies as the recovered barcodes with only 1–2 UMIs were likely missed by chance. Thus, the numbers of barcodes we found represent lower bounds of the barcode diversity in these libraries, and the shallow depth of sequencing could potentially explain the number of barcodes that were not matched to the libraries.

The RVΔG-4mCherry_CCS2_20nt_HM(B19G) library was used to prepare the RVΔG-4mCherry_CCS2_20nt_HM(EnvA) library, so we did not separately sequence the barcodes in the B19G coated library.

## M10. In situ sequencing – library preparation

In situ sequencing was performed as previously described for BARseq (*Sun et al., 2021*; *Chen et al., 2023*) and a detailed step-by-step protocol is provided at protocols.io (dx.doi.org/10.17504/protocols.io.n2bvj82q5gk5/v1). Briefly, brain sections were fixed in 4% paraformaldehyde in PBS for 1 hr, dehydrated through 70%, 85%, and 100% ethanol, and incubated in 100% ethanol for 1.5 hr at 4 °C. After rehydration in PBST (PBS and 0.5% Tween-20), we incubate in the reverse transcription mix (RT primers, 20 U/μL RevertAid H Minus M-MuLV reverse transcriptase, 500 μM dNTP, 0.2 μg/μL BSA, 1 U/μL RiboLock RNase Inhibitor, 1×RevertAid RT buffer) at 37 °C overnight. On the second day, we crosslink cDNA using BS(PEG)$_9$ (40 μL in 160 μL PBST) for 1 hr, neutralize the remaining crosslinker with 1 M Tris pH 8.0 for 30 min, and wash with PBST. We then incubate in non-gap-filling ligation mix (1×Ampligase buffer, padlock probe mix for endogenous genes, 0.5 U/μL Ampligase, 0.4 U/μL RNase H, 1 U/μL RiboLock RNase Inhibitor, additional 50 mM KCl, 20% formamide) for 30 min at 37 °C and 45 min at 45 °C, followed by incubating in the gap-filling ligation mix (same as the non-gap-filling mix with the rabies barcode padlock probe [XCAI5] as the only padlock probe, and with 50 μM dNTP, 0.2 U/μL Phusion DNA polymerase, and 5% glycerol) for 5 min at 37 °C and 45 min at 45 °C. After incubation, we wash the sample with PBST, hybridize with 1 μM RCA primer (XC1417) for 10 mins in 2×SSC with10% formamide, wash twice in 2×SSC with10% formamide and twice in PBST, then incubate in the RCA mix (1 U/μL phi29 DNA polymerase, 1 x phi29 polymerase buffer, 0.25 mM dNTP, 0.2 μg/μL BSA, 5% glycerol (extra of those from the enzymes), 125 μM aminoallyl dUTP) overnight at room temperature. On the third day we crosslink and neutralize additional crosslinkers as performed on the second day.

In the reverse transcription mix, the RT primers included 50 μM (final concentration) random primer (XC2757) and 2 μM each of primers against the rabies barcode flanking regions (XCAI6 and XCAI7). For the monosynaptic tracing experiments, the RT primers additionally included 2 μM of primers for the rabies glycoprotein (XCAI63 – XCAI76). All RT primers contain amine groups at the 5' end, which allow crosslinking on the second day. In the retrograde tracing experiments, the non-gap-filling padlock probe mix included all padlock probes for endogenous genes. In the monosynaptic tracing experiments, the non-gap-filling padlock probe mix included all padlock probes for endogenous genes except *Slc30a3*, and additionally included padlocks for the rabies glycoprotein (XCAI77 – XCAI90).

## M11. In situ sequencing – sequencing and imaging

Sequencing was performed using Illumina MiSeq Reagent Nano Kit v2 (300-cycles) and imaged on a Nikon Ti2-E microscope with Crest xlight v3 spinning disk confocal, photometrics Kinetix camera, and Lumencor Celesta laser. All images were taken with a Nikon CFI S Plan Fluor LWD 20XC objective. For each sample, we first sequence seven sequencing cycles for genes, followed by one hybridization cycle, followed by 15 sequencing cycles for barcodes. In each imaging round, we took a z-stack of 21 images centered around the tissue with 1.5 μm step size at each field of view (FOV). Adjacent FOVs had an overlap of 24%. Detailed list of filters and lasers used for each imaging channel are listed in *Table 3*.

Detailed sequencing protocols are available at protocols.io (https://www.protocols.io/view/barseq-barcoded-rabies-cmppu5mn), and a brief description is provided here.

For sequencing the first gene cycles, we hybridized sequencing primer (YS220) in 2×SSC with10% formamide for 10 min at room temperature, wash twice in 2×SSC with10% formamide and twice in PBS2T (PBS with 2% Tween-20). We then incubate the sample in MiSeq Incorporation buffer at 60 °C for 3 min, wash in PBS2T, then incubate the sample in Iodoacetamide (9.3 mg vial in 2.5 mL PBS2T) at

**Table 3.** List of filters and lasers used for in situ sequencing.

*Filters*

| Main dichroic | Filter names |
| --- | --- |
| D1 | FF421/491/567/659/776-Di01 (Semrock) |
| D2 | ZT405/514/635rpc |
| D3 | FF421/491/572-Di01−25x36(Semrock) |

| Emission filters | |
| --- | --- |
| E1 | FF01-441/511/593/684/817(Semrock) |
| E2 | FF01-565/24(Semrock) |
| E4 | FF01-676/29(Semrock) |
| E5 | FF01-775/140(Semrock) |
| E7 | 69,401 m |
| E8 | ZET532/640 m |

*Imaging settings*

| Sequencing cycles | | |
| --- | --- | --- |
| Channel | Filter combinations | laser |
| G | D2/E2 | 520 |
| T | D1/E1 | 546 |
| A | D2/E4 | 638 |
| C | D2/E5 | 638 |
| DIC | D2/E5 | DIA |

| Hybridization cycle | | |
| --- | --- | --- |
| GFP | D3/E7 | 477 |
| YFP | D2/E2 | 520 |
| TxRed | D3/E7 | 546 |
| Cy5 | D2/E8 | 638 |
| DAPI | D1/E7 | 405 |
| DIC | D3/E7 | DIA |

60 °C for 3 min, wash once in PBS2T and twice in Incorporation buffer, incubate twice in IMS at 60 °C for 3 min, wash four times with PBS2T at 60 °C for 3 min, then wash and image in SRE.

For sequencing the first barcode cycle, we first strip the sequenced products with three 10 min incubations at 60 °C in 2×SSC with 60% formamide. We then wash with 2×SSC with10% formamide, hybridize the barcode sequencing primer (XCAI5) and proceed with sequencing in the same way as for the first gene cycle.

For subsequent sequencing cycles for both genes and barcodes, we wash twice in Incorporation buffer, incubate twice in CMS at 60 °C for 3 min, wash twice in Incorporation buffer, then proceed with incubation with iodoacetamide as performed for the first sequencing cycle.

For hybridization cycles, we strip the sequenced products with three 10 min incubations at 60 °C in 2×SSC with 60% formamide. We then wash with 2×SSC with 10% formamide, hybridize the hybridization probes (XC2758, XC2759, XC2760, YS221) for 10 min at room temperature, then wash twice with 2×SSC with 10% formamide, and incubate in PBST with 0.002 mg/mL DAPI for 5 min. We then change to SRE and image.

The control brain was sectioned to 50 μm and imaged using the DAPI channel, the T channel, and DIC channel settings.

## M12. In situ sequencing – data processing

In situ sequencing data was processed in MATLAB using custom scripts (see Data and Code Availability; *Chen et al., 2023*). In general, we processed each FOV separately, and only combined data from FOV after extracting rolony-level and cell-level data. We denoised the max projection images from each imaging stack using noise2void (*Krull et al., 2018*), fixed x-y shifts between channels, and corrected channel bleed through. We then registered all gene sequencing cycles to the first gene sequencing cycle using the sum of the signals from all four sequencing channels; we registered the hybridization cycle to the first gene sequencing cycle using the TxRed channel, which visualized all the sequenced gene rolonies; we registered all barcode sequencing cycles to the first barcode sequencing cycle using the sum of sequencing signals; and we registered the first barcode sequencing cycle to the first gene sequencing cycle using the brightfield image. We then performed cell segmentation with Cellpose (*Stringer et al., 2021*), using the sum of all four hybridization channels, which visualize all gene rolonies, as cytoplasmic signals and the DAPI channel in hybridization cycle as nuclear signals. Gene rolonies are decoded using Bardensr (*Chen et al., 2021*). Barcode rolonies were first identified by picking peaks in the first barcode cycle using both prominence and intensity thresholds. We then called the nucleotide at each cycle as the channel with the strongest signal of the four sequencing channels. Within each cell, barcodes within two mismatches away from other barcodes are error-corrected to the most abundant barcodes within the same cell. All FOVs were stitched using ImageJ to find the transformation matrix from each FOV to each slice. These transformation matrices were then applied to the position of cells and rolonies to find their positions in each slice. Cells that were imaged multiple times in the overlapping regions between neighboring FOVS were removed using an approach based on AABB collision detection (*Baraff and Witkin, 1992*).

We filtered out neurons with fewer than 20 counts/cell and 5 genes/cell, then clustered the cells in an iterative approach using Louvain clustering as described previously (*Chen et al., 2023*). Clusters were mapped to reference scRNA-seq data (*Tasic et al., 2018*) using SingleR (*Aran et al., 2019*). We registered slices to the Allen CCF v3 using a manual procedure described previously (*Chen et al., 2023*) based on QuickNII and Visualign (*Puchades et al., 2019*).

## M13. Multiplexed retrograde tracing analysis

To estimate an upper bound of the false positive rate for the retrograde tracing experiment using scRNA-seq, we used the definition of false positive rate: $FPR = FP/(FP + TN)$, where $FP$ is the number of false positive detections and $TN$ is the number of true negative detections. Because L5 ET and L6 CT neurons are the main neuronal types that project to the LGd (*Harris and Shepherd, 2015*; *Tasic et al., 2018*), we defined $TN$ as all sequenced inhibitory neurons, IT neurons and L5 NP neurons that did not project to the LGd (24 IT neurons, 1 L5 NP neurons, and 7 inhibitory neurons; total 32 neurons), and $FP$ as those that did project to the LGd (0 neurons). We did not include L6b neurons, because some L6b neurons form a continuum with L6 CT neurons in transcriptomics and morphology. It is thus unclear whether L6b neurons could project to the thalamus. Since no $FP$ was identified in this dataset, the true $FP$ likely lies between 0 and 1. We thus estimate that the upper bound of $FPR$ as $(1 + FP)/(FP + TN) = 1/32$, or 3.1%.

The analysis of the in situ sequencing experiment was performed in MATLAB. Cells were considered barcoded if it contained at least six reads of the same barcode, and the barcodes were matched to the two sequencing libraries allowing one mismatch. To analyze the spatial patterns of the somas of projection neurons, we first divided the cortex into 267 'cubelets'. On each slice, the cubelets were drawn so that they spanned all layers of the cortex and had roughly similar width (~100 μm) along the outer surface of the cortex along the mediolateral axis. These cubelets largely covered cortical areas VISp, VISl, ECT, and TEa. We did not draw over RSP because the curvature of the cortex at RSP makes it difficult to define cubelets in this area. In each cubelet $c$, we denoted the projection probability for each cell type $t$ as $P_{ct}$. Let $M = mean_{c \in cubelets, \ t \in types}(P_{ct})$ be the mean projection probability across all cell types and cubelets, and $TV = \sum_{c \in cubelets, \ t \in types}(P_{ct} - M)^2$ be the total variance. We then computed the variance explained by areas, subclasses, types, and types with areas as follows:

$$VE_{area} = \sum_{a \in area}\left(mean_{c \in a, t \in types}(P_{ct}) - M\right)^2 /TV$$

$$VE_{subclass} = \sum_{h \in subclass} \left( mean_{c \in cubelets, t \in h} \left( P_{ct} \right) - M \right)^2 / TV$$

$$VE_{type} = \sum_{t \in type} \left( mean_{c \in cubelets} \left( P_{ct} \right) - M \right)^2 / TV$$

$$VE_{type,area} = \sum_{t \in type, a \in area} \left( mean_{c \in a} \left( P_{ct} \right) - M \right)^2 / TV$$

To calculate the variance explained by types and shuffled areas, we first shuffled the area labels of all 267 cubelets, then followed the equations about for $VE_{type,area}$ .

## M14. Barcoded transsynaptic labeling analysis

For the scRNA-seq based transsynaptic labeling experiment, barcode-level analyses, including Hamming distance and matching to barcode library, were performed in MATLAB.

For the in situ sequencing experiments, we filtered barcodes based on linguistic sequence complexity (*Trifonov, 1990*) and barcode counts per cell. For a barcode with length $M$, the linguistic sequence complexity $C$ is defined as the product of the vocabulary-usage measures of $n$-grams, $U_n$, for all possible $ns$:

$$C = \prod_{n=1}^{M} U_n$$

where $U_n$ is defined as the ratio between the number of unique n-grams that was observed and the minimum between all possible n-grams and the total number of n-grams in that sequence. We first curated barcoded cells manually to determine appropriate thresholds. We generated 100×100 pixels crops of images of all barcode sequencing cycles, the first gene sequencing cycle, the hybridization cycle channels, and cell segmentation mask for 262 cells with varying barcode counts per cell and barcode complexity. The images were visually inspected to determine whether a barcode with the called barcode is present in the cell to determine the thresholds for complexity and barcode counts that removed wrong barcodes. These barcodes were dominated by incomplete cells at the borders of an FOV (low complexity), autofluorescence background (low complexity), and wrong basecalls due to overlapping barcode rolonies (low counts). To determine the source cells, we additionally required cells to have at least two read counts of the rabies glycoprotein. This threshold was chosen so that most glycoprotein positive cells were around the injection sites and were concentrated in layer 5, which are enriched in corticostriatal projection neurons.

To estimate the fraction of barcodes that were observed in source cells, we only focused on barcodes that had sufficient reads per cell (≥10 reads per cell) and were in networks that were large enough to exclude most no-source networks and lost-source networks (≥12 cells per barcode). We then asked what fraction of those barcodes were found in source cells, which were defined by having a barcode with at least X reads per cell. Because we knew based on manual inspection that all barcodes with at least ten reads per cell were real, but only some barcodes with between three to ten reads per cell were real, we reasoned that thresholding at X=10 would likely underestimate the number of barcodes that were found in source cells, whereas thresholding at X=3 would overestimate the number of barcodes that were found in source cells. Estimates using these two thresholds thus provided the upper and lower bound of the true rate at which barcodes were observed in source cells.

To estimate the number of independent infection events that achieved the number of barcodes in source cells observed in the experiment, we calculated the posterior probability, P(N/M), of using N independent infection events to achieve $M = 59$ unique barcodes using Bayes' theorem:

$$P\left(N|M\right) = \frac{P\left(M|N\right) \times P\left(N\right)}{P\left(M\right)}$$

where P(N/M) is the probability of drawing M unique barcodes after N infections based on the barcode frequency in the virus library, P(N) is the probability of having N infections (uniform distribution), and P(M) is the probability of drawing M unique barcodes overall:

$$P\left(M\right) = \sum_{N=M}^{C} P\left(M|N\right) \times P\left(N\right)$$

where $C = 126$ is the number of unique barcode-source cell combinations that was observed.

To find biases in converging outputs between cell types, we first removed all barcodes with fewer than five cells, which resulted in 153 barcodes. This lower boundary allowed us to filter out most no-source networks. We then estimated the number of independent infection events and the maximum barcode frequency to achieve 95% single infection. We randomly drew barcodes from the barcode library based on their frequencies until we obtained the same number of barcodes as in the in situ dataset. Because the library was sequenced only to 82%, we considered the remaining 18% of barcodes to be all non-repetitive for the purpose of this analysis. We repeated this process 10,000 times and used the median numbers as estimate for the total number of infection events and the barcode frequency threshold for the subsequent analysis. For each barcode, we considered all pair-wise combinations of cells that share that barcode to have converging outputs and accumulated the number of converging outputs across all barcodes for each combination of cortical subclasses. To estimate the biases compared to the expected number of converging outputs, we calculated the expected number of converging outputs as the square of cell counts for each cortical subclass, normalized so that the total number of connections is the same as the data. To estimate p values, we shuffled the identity of connected cells and recalculated the number of converging outputs for each pair of cortical subclasses. We repeated this process 10,000 times and used two times the fraction of iterations with more extreme values of converging outputs as the p values.

Estimating the barcode frequency threshold required knowing the distribution of barcodes in the barcode library. Because the library sequencing was very shallow (most UMIs had a single read), for barcodes with very few UMIs, we could not distinguish whether those barcodes were real barcode/UMI pairs or whether they resulted from PCR errors (**Kebschull and Zador, 2015**). Furthermore, many barcodes that had low frequencies in the library (which were ideal for unique labeling) were likely missed from the barcode sequencing. Thus, we could not get an accurate estimate of the distribution for the rare barcodes. However, since double labeling was more likely to occur to barcodes that were more abundant, we only needed to know the distribution of abundant barcodes to find a frequency threshold that can ensure a small number of double labeling events. Thus, we thus used the most abundant 1,820 barcodes (out of 13,211 barcodes observed) in the RVΔG-4mCherry_CCS2_20nt_HM (EnvA) library when estimating the frequency threshold; these 1,820 barcodes each had at least 21 UMI counts and constituted 81.9% of all barcode molecules in this library.

Detailed analysis scripts are provided on Mendeley Data (see **Data and code availability**).

## Materials availability statement

Samples of novel plasmids, including the barcoded rabies virus genome plasmids, are available from the authors on request.

## Acknowledgements

The authors thank Yoav Ben-Simon, Ian Peikon, Justus Kebschull, Anthony Zador, Mara Rue for discussion. This work was supported by the National Institute of Health (1DP2MH132940 to XC) and by seed grants from the James W and Patricia T Poitras Fund and the Charles S Camplan Fund to IRW. The content is solely the responsibility of the authors and does not necessarily represent the official views of the National Institutes of Health.

## Additional information

### Competing interests

Ian Wickersham: IRW is a consultant for Monosynaptix, LLC, advising on design of neuroscientific experiments. The other authors declare that no competing interests exist.

## Funding

| Funder | Grant reference number | Author |
|---|---|---|
| National Institutes of Health | 1DP2MH132940 | Xiaoyin Chen |
| James W. and Patricia T. Poitras Fund | | Ian Wickersham |
| Charles S. Camplan Fund | | Ian Wickersham |

The funders had no role in study design, data collection and interpretation, or the decision to submit the work for publication.

## Author contributions

Aixin Zhang, Cindy TJ van Velthoven, Data curation, Software, Formal analysis, Investigation, Visualization; Lei Jin, Resources, Formal analysis, Investigation, Methodology, Writing – original draft, Writing – review and editing; Shenqin Yao, Data curation, Software, Formal analysis, Investigation, Visualization, Methodology, Writing – original draft, Writing – review and editing; Makoto Matsuyama, Heather Anne Sullivan, Resources, Methodology; Na Sun, Software, Formal analysis; Manolis Kellis, Formal analysis, Supervision; Bosiljka Tasic, Conceptualization, Data curation, Supervision, Writing – original draft, Writing – review and editing; Ian Wickersham, Conceptualization, Resources, Formal analysis, Supervision, Funding acquisition, Investigation, Visualization, Methodology, Writing – original draft, Writing – review and editing; Xiaoyin Chen, Conceptualization, Data curation, Software, Formal analysis, Supervision, Funding acquisition, Investigation, Visualization, Methodology, Writing – original draft, Writing – review and editing

## Author ORCIDs

Lei Jin ⓘ https://orcid.org/0000-0002-7264-4409
Shenqin Yao ⓘ http://orcid.org/0000-0003-2992-4752
Cindy TJ van Velthoven ⓘ http://orcid.org/0000-0001-5120-4546
Bosiljka Tasic ⓘ http://orcid.org/0000-0002-6861-4506
Ian Wickersham ⓘ https://orcid.org/0000-0003-0389-5324
Xiaoyin Chen ⓘ http://orcid.org/0000-0002-2807-6125

## Ethics

Animal handling and surgery were conducted according to protocols approved by the Institutional Animal Care and Use Committee (IACUC) of the Allen Institute for Brain Science (protocol number 2106 and 2201) and protocol approved by the MIT Committee on Animal Care (protocol number 2303000498).

Reviewer #1 (Public Review): https://doi.org/10.7554/eLife.87866.3.sa1
Reviewer #2 (Public Review): https://doi.org/10.7554/eLife.87866.3.sa2
Reviewer #3 (Public Review): https://doi.org/10.7554/eLife.87866.3.sa3
Author Response https://doi.org/10.7554/eLife.87866.3.sa4

# Additional files

## Supplementary files

- Supplementary file 1. Metadata of animals used in this study.
- Supplementary file 2. Viruses used in this study.
- Supplementary file 3. Gene panel and list of oligos used.
- MDAR checklist

## Data availability

Single-cell sequencing data were deposited at the Neuroscience Multi-omic Data Archive (NeMO). Sequences of the barcoded rabies library were deposited at Sequence Read Archive (SRA SRR23310758, SRR23310757, and SRR23310756). Processed in situ sequencing data, processing scripts, analysis scripts, and data manifest for the NeMO data are deposited at Mendeley Data (https://

The following datasets were generated:

| Author(s) | Year | Dataset title | Dataset URL | Database and Identifier |
|---|---|---|---|---|
| Chen X, Jin L, Wickersham I, Yao S, Tasic B, von Velthoven C, Zhang A | 2023 | Rabies virus-based barcoded neuroanatomy resolved by single-cell RNA and in situ sequencing | https://doi.org/10. 17632/dctm2htrf5.2 | Mendeley Data, 10.17632/ dctm2htrf5.2 |
| Chen X, Jin L, Wickersham I, Zhang A | 2023 | Multiplexed retrograde labeling using barcoded rabies virus with Barseq: 602548 | https://download. brainimagelibrary. org/66/8c/ 668cc170c96d5eaf/ 20220127_bcrabies_ M1_602546_602548/ 602548 | Brain Image Library, ace-den-gap |
| Chen X, Jin L, Wickersham I, Zhang A | 2023 | Multiplexed monosynaptic tracing using barcoded rabies virus with Barseq | https://download. brainimagelibrary. org/3e/75/ 3e75e706f8d4407d | Brain Image Library, ace-cry-ash |
| Jin L, Wickersham I | 2023 | RNA-seq of barcodes in barcoded rabies virus | https://www.ncbi.nlm. nih.gov/sra/?term= SRR23310758 | NCBI Sequence Read Archive, SRR23310758 |
| Jin L, Wickersham I | 2023 | RNA-seq of barcodes in barcoded rabies virus | https://www.ncbi.nlm. nih.gov/sra/?term= SRR23310756 | NCBI Sequence Read Archive, SRR23310756 |
| Jin L, Wickersham I | 2023 | RNA-seq of barcodes in barcoded rabies virus | https://www.ncbi.nlm. nih.gov/sra/?term= SRR23310757 | NCBI Sequence Read Archive, SRR23310757 |

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

## Appendix 1

We tested whether scRNA-seq was sufficient to map single-cell connectivity using barcoded transsynaptic labeling. We first injected a Cre-dependent helper AAV virus combination (*Liu et al., 2017*; *Lavin et al., 2019*; *Lavin et al., 2020*) to express TVA (co-expressed with EGFP) and the rabies glycoprotein (co-expressed with mTagBFP2) in Cre-defined populations of cells in VISp (three Sst-IRES-Cre animals and one Vip-IRES-Cre animal). After 14 days, we injected a barcoded ΔG rabies virus library expressing mCherry and pseudotyped with EnvA for selective infection of TVA-expressing neurons. This library contained the same barcodes as the CCS-containing library used for retrograde labeling, with at least 13,211 barcodes. After another seven days, we dissected out the visual cortex, isolated both mTagBFP2+/mCherry +neurons and mTagBFP2-/mCherry +neurons using fluorescence-activated cell sorting (FACS), and performed scRNA-seq on both populations using SMART-Seq v4 (total 295 cells from four animals; *Table 1*; *Figure 5—figure supplement 1A*). After quality control, we identified 195 potential presynaptic neurons (defined as expressing rabies viral genes, but not transcripts from both helper AAV-viruses) and 37 potential source neurons (defined as expressing both helper AAV transcripts and the other rabies viral genes) (*Figure 5—figure supplement 1B*). Consistent with the Cre lines used, the source neurons were predominantly Sst neurons in the three Sst-IRES-Cre animals, and predominantly Vip neurons in the Vip-IRES-Cre animal (*Figure 5—figure supplement 1C*).

To assess the accuracy of barcode sequencing using scRNA-seq, we compared the distribution of the minimum Hamming distance between barcode pairs in our dataset or a random set of 20-nt barcodes of similar size (*Figure 5—figure supplement 1D*). Whereas no random barcodes had minimum Hamming distance of 5 or less, the distribution of minimum Hamming distance for barcodes had two peaks at 1 and 4, respectively. These two peaks likely correspond to sequencing errors and errors resulting from having multiple barcodes per cell. We thus corrected barcodes by allowing up to five mismatches (Materials and methods).

We then identified and removed double-labeled networks from the barcoded cells. Double-labeled networks should be largely restricted to barcodes that were abundant in the virus library. We thus estimated the prevalence of these networks based on barcode distribution across different animals: if a barcode was found in multiple animals, then it would be unlikely that it labeled a single source cell in one of the animals. Of 20 unique barcodes in all cells, six barcodes were found in only one animal, whereas eight of the 14 remaining barcodes were found in all four animals (*Figure 5—figure supplement 1E*). Consistent with the hypothesis that these barcodes were abundant in the virus library, the frequencies of barcodes found in all four animals ranged from 0.27% to 0.03% (median 0.08%), whereas the frequencies of barcodes that were found in only one animal were all <0.01% with median at $4 \times 10^{-5}$ ($P = 7 \times 10^{-4}$ compared to the frequency of barcodes that were found in all animals using rank sum test; *Figure 5—figure supplement 1F*). Of the six barcodes that were found in only one animal, five were found in a single neuron each, and the remaining barcode was found only in one source neuron and one non-source (i.e., not expressing G) neuron. Thus, after removing barcodes that likely generated double-labeled networks, the remaining barcoded cells were too few to infer connections in these four animals. We attributed this to cell loss from dissociation and FACS that is inherent to the scRNA-seq pipeline.

## Appendix 2

In conventional rabies monosynaptic tracing experiments, source cells are usually determined by fluorescent proteins that co-express with the rabies glycoprotein. The in situ sequencing procedure we used did not robustly preserve fluorescence in the tissue, so we detected the transcripts of the rabies glycoprotein to identify source cells. Because we did not know how RNA transcript levels relate to visible co-expression of fluorescent proteins, we varied the threshold above which we considered a cell to express the glycoprotein and examined the composition of cell types within this population (*Figure 5—figure supplement 2B*). We reasoned that since corticostriatal neurons mainly consisted of deep layer excitatory neurons, an equivalent threshold in the glycoprotein transcript should result in cells that were mostly deep-layer IT neurons and L5 ET neuron, but not L6 CT neurons or inhibitory neurons. At a low threshold, the labeled cells contained a large portion of inhibitory neurons; as we increased the threshold, inhibitory neurons were preferentially removed compared to excitatory neurons. With a threshold of 12 glycoprotein reads per cell, the labeled cells were predominantly L5 ET neurons, and L6 IT neurons. Based on these results, we considered a glycoprotein read count of about 12 as equivalent to having visible fluorescence in a conventional experiment.

Although neurons with fewer than 12 glycoprotein reads per cell likely did not transmit the rabies virus efficiently, they may still contribute to smaller barcode-sharing networks. Because these barcode-sharing networks could either be connected-source networks (i.e., if a source cell with few glycoprotein reads was presynaptic to another source cell) or double-labeled networks (i.e., if a source cell with few glycoprotein reads was independently infected with the same barcode as another source cell), removing these cells from the population of source cells would convert these networks into single-source networks and confound the interpretation of synaptic connectivity at cellular resolution. Using a lower threshold would potentially include false positive source cells that did not pass the barcodes to other neurons. Such neurons, however, would not contribute to barcode-sharing networks. Thus, to ensure the single-cell resolution in our analyses, we used a conservative threshold of two glycoprotein reads per cell to identify potential source cells. Using this threshold, potential source cells remained largely mostly in layer 5, suggesting that corticostriatal neurons remained enriched using this conservative threshold (*Figure 5—figure supplement 2C*).

To assess whether the labeling patterns in the barcoded trans-synaptic labeling experiment was reasonable, we performed a control experiment in an animal that was injected in parallel as the barcoded trans-synaptic labeling experiment (*Figure 5B*). We then examined fluorescent protein expression in this animal (*Figure 5—figure supplement 2D*). In a coronal section that included the injection site (*Figure 5—figure supplement 2D*, left), neurons were broadly infected by rabies virus and expressed mCherry from the rabies genome. Sparser labeling was seen in the striatum and in medial cortical areas (motor cortex and anterior cingulate cortex). In a second coronal section that was about 1 mm posterior to the injection site (*Figure 5—figure supplement 2D*, right), more labeling was seen in the auditory cortex and areas of the thalamus that roughly corresponded to posterior complex of the thalamus and the ventral posteromedial nucleus (*Figure 5—figure supplement 2D*, inset), which were known to project to the primary somatosensory cortex (*Harris et al., 2019*). Thus, the labeling pattern was largely consistent with the input patterns of the primary somatosensory cortex (*Harris et al., 2019*).

We next examined the spatial distribution of potential source cells. Consistent with the fact that L5 neurons are enriched for corticostriatal projections, most BFP-expressing cells were found in L5 (*Figure 5—figure supplement 2E*, middle). The area that most BFP-expressing cells were found, however, only partially overlapped with the region with the bulk of mCherry-expressing rabies-infected cells, which appeared to be centered around the injection site in the cortex based on the depression on the cortical surface (*Figure 5—figure supplement 2E*, right). Thus, the spatial pattern of BFP expression was consistent with retrograde labeling from the striatum, but they only partially overlapped with the rabies virus injection in the cortex. This partial overlap likely contributed to the sparseness of neurons that were trans-synaptically labeled in distant areas (e.g., in the thalamus). Furthermore, because presynaptic neurons are usually centered around the source cells in the same cortical area (*Wertz et al., 2015*; *Yao et al., 2023*), we speculate that a large fraction of rabies-infected neurons likely resulted from direct infections that were independent from source cells (i.e., forming no-source networks). These analyses are consistent with the high number of no-source networks observed in the barcoded experiment.

