## [Editor Report · eLife assessment]

This study presents an **important** tool for tracking the connectivity of neurons in mouse and potentially other mammals using a combined approach of barcoded rabies virus libraries and spatial transcriptomics. The data supporting the technique are **convincing**, the validation against known anatomical knowledge is rigorous, and the authors advance the techniques by combing them in vivo. Overall, this is a very good paper describing a technique for tracking neural circuits.

---

## [Referee Report · Reviewer #1 (Public Review)]

In this preprint, Zhang et al. describe a new tool for mapping the connectivity of mouse neurons. Essentially, the tool leverages the known peculiar infection capabilities of Rabies virus: once injected into a specific site in the brain, this virus has the capability to "walk upstream" the neural circuits, both within cells and across cells: on one hand, the virus can enter from a nerve terminal and infect retrogradely the cell body of the same cell (retrograde transport). On the other hand, the virus can also sometimes spread to the presynaptic partners of the initial target cells, via retrograde viral transmission.

Similarly to previously published approaches with other viruses, the authors engineer a complex library of viral variants, each carrying a unique sequence ('barcode'), so they can uniquely label and distinguish independent infection events and their specific presynaptic connections, and show that it is possible to read these barcodes in-situ, producing spatial connectivity maps. They also show that it is possible to read these barcodes together with endogenous mRNAs, and that this allows spatial mapping of cell types together with anatomical connectivity.

The main novelty of this work lies in the combined use of rabies virus for retrograde labeling together with barcoding and in-situ readout. Previous studies had used rabies virus for retrograde labeling, albeit with low multiplexing capabilities, so only a handful of circuits could be traced at the same time. Other studies had instead used barcoded viral libraries for connectivity mapping, but mostly focused on the use of different viruses for labeling individual projections (anterograde tracing) and never used a retrograde-infective virus.

The authors creatively merge these two bits of technology into a powerful genetic tool, and extensively and convincingly validate its performance against known anatomical knowledge. The authors also do a very good job at highlighting and discussing potential points of failure in the methods.

Unresolved questions, which more broadly affect also other viral-labeling methods, are for example how to deal with uneven tropism (ie. if the virus is unable or inefficient in infecting some specific parts of the brain), or how to prevent the cytotoxicity induced by the high levels of viral replication and expression, which will tend to produce "no source networks", neural circuits whose initial cell can't be identified because it's dead. This last point is particularly relevant for in-situ based approaches: while high expression levels are desirable for the particular barcode detection chemistry the authors chose to use (gap-filling), they are also potentially detrimental for cell survival, and risk producing extensive cell death (which indeed the authors single out as a detectable pitfall in their analysis). This is likely to be one the major optimisation space for future implementations of this barcoding approach.

Overall the paper is well balanced, the data are well presented and the conclusions are strongly supported by the data. Impact-wise, the method is definitely going to be very useful for the neurobiology community.

---

## [Referee Report · Reviewer #2 (Public Review)]

Although the trans-synaptic tracing method mediated by the rabies virus (RV) has been widely utilized to infer input connectivity across the brain to a genetically defined population in mice, the analysis of labeled pre-synaptic neurons in terms of cell-type has been primarily reliant on classical low-throughput histochemical techniques. In this study, the authors made a significant advance toward high-throughput transcriptomic (TC) cell typing by both dissociated single-cell RNAseq and the spatial TC method known as BARseq to decode a vast array of molecularly labeled ("barcoded") RV vector library. First, they demonstrated that a barcoded RV vector can be employed as a simple retrograde tracer akin to AAVretro. Second, they provided a theoretical classification of neural networks at the single-cell resolution that can be attained through barcoded-RV and concluded that the identification of the vast majority (ideally 100%) of starter cells (the origin of RV-based trans-synaptic tracing) is essential for the inference of single-cell resolution neural connectivity. Taking this into consideration, the authors opted for the BARseq-based spatial TC that could, in principle, capture all the starter cells. Finally, they demonstrated the proof-of-concept in the somatosensory cortex, including infrared connectivity from 381 putative pre-synaptic partners to 31 uniquely barcoded-starter cells, as well as many insightful estimations of input convergence at the cell-type resolution in vivo. Collectively, this work will establish a cornerstone for future advancements in rabies-barcode technology.

This revised version incorporates imaging data to assess the stringency of identifying the starter cells in comparison with conventional protein-based detection methods. Additionally, it encompasses insightful discussions concerning potential limitations and offers perspectives on future improvements. The method section is systematically subdivided with subsection numbers, facilitating the cross-referencing of the corresponding sections in the main text and figure legends. I posit that adopting this stylistic approach as the standard for manuscripts delineating innovative methodological strides would be prudent. The clarity of the figure legends has been significantly enhanced, contributing to a more accessible understanding of the figure panels. In sum, this manuscript is articulate and thorough, epitomizing scientific rigor.

---

## [Referee Report · Reviewer #3 (Public Review)]

The manuscript by Zhang and colleagues attempts to combine genetically barcoded rabies viruses with spatial transcriptomics in order to genetically identify connected pairs. The major shortcoming with the application of a barcoded rabies virus, as reported by 2 groups prior, is that with the high dropout rate inherent in single cell procedures, it is difficult to definitively identify connected pairs. By combining the two methods, they are able to establish a platform for doing that, and provide insight into connectivity, as well as pros and cons of their method, which is well thought out and balanced.

The authors did a nice job of addressing my comments which mainly centered around the presentation of data, specificity, and wording.

---

## [Author Response]

The following is the authors’ response to the original reviews.

**eLife assessment**
This important study combines genetically barcoded rabies viruses with spatial transcriptomics in vivo in the mouse brain to decode connectivity of neural circuits. The data generated by the combination of these approaches in this new way is mostly convincing as the authors provide validation and proof-of-concept that the approach can be successful. While this new combination of established techniques has promise for elucidating brain connectivity, there are still some nuances and caveats to the interpretations of the results that are lacking especially with regards to noting unexpected barcodes either due to unexpected/novel connections or unexpected rabies spread.

In this revised manuscript, we added a new control experiment and additional analyses to address two main questions from the reviewers: (1) How the threshold of glycoprotein transcript counts used to identify source cells was determined, and (2) whether the limited long-range labeling was expected in the trans-synaptic experiment. The new experiments and analyses validated the distribution of source cells and presynaptic cells observed in the original barcoded transsynaptic tracing experiment and validated the choice of the threshold of glycoprotein transcripts. As the reviewers suggested, we also included additional discussion on how future experiments can improve upon this study, including strategies to improve source cell survival and minimizing viral infection caused by leaky expression of TVA. We also provided additional clarification on the analyses for both the retrograde labeling experiment and the trans-synaptic tracing experiment. We modified the Results and Discussion sections on the trans-synaptic tracing experiment to improve clarity to general readers. Detailed changes to address specific comments by reviewers are included below.

**Public Reviews:**

**Reviewer #1 (Public Review):**
In this preprint, Zhang et al. describe a new tool for mapping the connectivity of mouse neurons. Essentially, the tool leverages the known peculiar infection capabilities of Rabies virus: once injected into a specific site in the brain, this virus has the capability to "walk upstream" the neural circuits, both within cells and across cells: on one hand, the virus can enter from a nerve terminal and infect retrogradely the cell body of the same cell (retrograde transport). On the other hand, the virus can also spread to the presynaptic partners of the initial target cells, via retrograde viral transmission.Similarly to previously published approaches with other viruses, the authors engineer a complex library of viral variants, each carrying a unique sequence ('barcode'), so they can uniquely label and distinguish independent infection events and their specific presynaptic connections, and show that it is possible to read these barcodes in-situ, producing spatial connectivity maps. They also show that it is possible to read these barcodes together with endogenous mRNAs, and that this allows spatial mapping of cell types together with anatomical connectivity.The main novelty of this work lies in the combined use of rabies virus for retrograde labeling together with barcoding and in-situ readout. Previous studies had used rabies virus for retrograde labeling, albeit with low multiplexing capabilities, so only a handful of circuits could be traced at the same time. Other studies had instead used barcoded viral libraries for connectivity mapping, but mostly focused on the use of different viruses for labeling individual projections (anterograde tracing) and never used a retrograde-infective virus.The authors creatively merge these two bits of technology into a powerful genetic tool, and extensively and convincingly validate its performance against known anatomical knowledge. The authors also do a very good job at highlighting and discussing potential points of failure in the methods.

We thank the reviewer for the enthusiastic comments.

Unresolved questions, which more broadly affect also other viral-labeling methods, are for example how to deal with uneven tropism (ie. if the virus is unable or inefficient in infecting some specific parts of the brain), or how to prevent the cytotoxicity induced by the high levels of viral replication and expression, which will tend to produce "no source networks", neural circuits whose initial cell can't be identified because it's dead. This last point is particularly relevant for in-situ based approaches: while high expression levels are desirable for the particular barcode detection chemistry the authors chose to use (gap-filling), they are also potentially detrimental for cell survival, and risk producing extensive cell death (which indeed the authors single out as a detectable pitfall in their analysis). This is likely to be one of the major optimisation challenges for future implementations of these types of barcoding approaches.

As the reviewer suggested, we included additional discussion about tropism and cytotoxicity in the revised Discussion. Our sensitivity for barcode detection is sufficient, since we estimated (based on manual proofreading) that most barcoded neurons had more than ten counts of a barcode in the trans-synaptic tracing experiment. The high sensitivity may potentially allow us to adapt next-generation rabies virus with low replication, such as the third generation ΔL rabies virus (Jin et al, 2022, biorxiv) in future optimizations.

Overall the paper is well balanced, the data are well presented and the conclusions are strongly supported by the data. Impact-wise, the method is definitely going to be useful for the neurobiology research community.

We thank the reviewer for her/his enthusiasm.

**Reviewer #2 (Public Review):**

Although the trans-synaptic tracing method mediated by the rabies virus (RV) has been widely utilized to infer input connectivity across the brain to a genetically defined population in mice, the analysis of labeled pre-synaptic neurons in terms of cell-type has been primarily reliant on classical low-throughput histochemical techniques. In this study, the authors made a significant advance toward high-throughput transcriptomic (TC) cell typing by both dissociated single-cell RNAseq and the spatial TC method known as BARseq to decode a vast array of molecularly labeled ("barcoded") RV vector library. First, they demonstrated that a barcoded-RV vector can be employed as a simple retrograde tracer akin to AAVretro. Second, they provided a theoretical classification of neural networks at the single-cell resolution that can be attained through barcoded-RV and concluded that the identification of the vast majority (ideally 100%) of starter cells (the origin of RV-based trans-synaptic tracing) is essential for the inference of single-cell resolution neural connectivity. Taking this into consideration, the authors opted for the BARseq-based spatial TC that could, in principle, capture all the starter cells. Finally, they demonstrated the proof-of-concept in the somatosensory cortex, including infrared connectivity from 381 putative pre-synaptic partners to 31 uniquely barcoded-starter cells, as well as many insightful estimations of input convergence at the cell-type resolution in vivo. While the manuscript encompasses significant technical and theoretical advances, it may be challenging for the general readers of eLife to comprehend. The following comments are offered to enhance the manuscript's clarity and readability.

We modified the Results and Discussion sections on the trans-synaptic tracing experiment to improve clarity to general readers. We separated out the theoretical discussion about barcode sharing networks as a separate subsection, explicitly stated the rationale of how different barcode sharing networks are distinguished in the in situ trans-synaptic tracing experiment, and added additional discussion on future optimizations. Detailed descriptions are provided below.

Major points:1. I find it difficult to comprehend the rationale behind labeling inhibitory neurons in the VISp through long-distance retrograde labeling from the VISal or Thalamus (Fig. 2F, I and Fig. S3) since long-distance projectors in the cortex are nearly 100% excitatory neurons. It is also unclear why such a large number of inhibitory neurons was labeled at a long distance through RV vector injections into the RSP/SC or VISal (Fig. 3K). Furthermore, a significant number of inhibitory starter cells in the somatosensory cortex was generated based on their projection to the striatum (Fig. 5H), which is unexpected given our current understanding of the cortico-striatum projections.

The labeling of inhibitory neurons can be explained by several factors in the three different experiments.

(1) In the scRNAseq-based retrograde labeling experiment (Fig. 2 and Fig. S3), the injection site VISal is adjacent to VISp. Because we dissected VISp for single-cell RNAseq, we may find labeled inhibitory neurons at the VISp border that extend short axons into VISal. We explained this in the revised Results.

(2) In the in situ sequencing-based retrograde labeling experiment (Fig. 3,4), the proximity between the two injection sites VISal and RSP/SC, and the sequenced areas (which included not only VISp but also RSP) could also contribute to labeling through local axons of inhibitory neurons. Furthermore, because we also sequenced midbrain regions, inhibitory neurons in the superior colliculus could pick up the barcodes through local axons. We included an explanation of this in the revised Results.

(3) In the trans-synaptic tracing experiment, we speculate that low level leaky expression from the TREtight promoter led to non-Cre-dependent expression in many neurons. To test this hypothesis, we first performed a control injection in which we saw that the fluorescent protein expression were indeed restricted to layer 5, as expected from corticostriatal labeling. Based on the labeling pattern, we estimated that about 12 copies of the glycoprotein transcript per cell would likely be needed to achieve fluorescent protein expression. Since many source cells in our experiment were below this threshold, these results support the hypothesis that the majority of source cells with low level expression of the glycoprotein were likely Cre-independent. Because these cells could still contribute to barcode sharing networks, we could not exclude them as in a conventional bulk trans-synaptic tracing experiment. In future experiments, we can potentially reduce this population by improving the helper AAV viruses used to express TVA and the glycoprotein. We included this explanation in Results and more detailed analysis in Supplementary Note 2, and discussed potential future optimizations in the Discussion.This new analysis in Supplementary Note 2 is also related to the Reviewer’s question regarding the threshold used for determining source cells (see below).

1. It is unclear as to why the authors did not perform an analysis of the barcodes in Fig. 2. Given that the primary objective of this manuscript is to evaluate the effectiveness of multiplexing barcoded technology in RV vectors, I would strongly recommend that the authors provide a detailed description of the barcode data here, including any technical difficulties or limitations encountered, which will be of great value in the future design of RV-barcode technologies. In case the barcode data are not included in Fig. 2, I would suggest that the authors consider excluding Fig. 2 and Fig. S1-S3 in their entirety from the manuscript to enhance its readability for general readers.

In the single-cell RNAseq-based retrograde tracing, all barcodes recovered matched to known barcodes in the corresponding library. We included a short description of these results in the revised manuscript.

1. Regarding the trans-synaptic tracing utilizing a barcoded RV vector in conjunction with BARseq decoding (Fig. 5), which is the core of this manuscript, I have a few specific questions/comments. First, the rationale behind defining cells with only two rolonies counts of rabies glycoprotein (RG) as starter cells is unclear. Why did the authors not analyze the sample based on the colocalization of GFP (from the AAV) and mCherry (from the RV) proteins, which is a conventional method to define starter cells? If this approach is technically difficult, the authors could provide an independent histochemical assessment of the detection stringency of GFP positive cells based on two or more colonies of RG.

In situ sequencing does not preserve fluorescent protein signals, so we used transcript counts to determine which cells expressed the glycoprotein. We have added new analyses in the Results and in Supplementary Note 2 to determine the transcript counts that were equivalent to cells that had detectable BFP expression. We found that BFP expression is equivalent to ~12 counts of the glycoprotein transcript per cell, which is much higher than the threshold we used. However, we could not solely rely on this estimate to define the source cells, because cells that had lower expression of the glycoprotein (possibly from leaky Cre-independent expression) may still pass the barcodes to presynaptic cells. This can lead to an underestimation of double-labeled and connected-source networks and an overestimation of single-source networks and can obscure synaptic connectivity at the cellular resolution. We thus used a very conservative threshold of two transcripts in the analysis. This conservative threshold will likely overestimate the number of source cells that shared barcodes and underestimate the number of single-source networks. Since this is a first study of barcoded transsynaptic tracing in vivo, we chose to err on the conservative side to make sure that the subsequent analysis has single-cell resolution. Future characterization and optimization may lead to a better threshold to fully utilize data.

Second, it is difficult to interpret the proportion of the 2,914 barcoded cells that were linked to barcoded starter cells (single-source, double-labeled, or connected-source) and those that remained orphan (no-source or lost-source). A simple table or bar graph representation would be helpful. The abundance of the no-source network (resulting from Cre-independent initial infection of the RV vector) can be estimated in independent negative control experiments that omit either Cre injection or AAV-RG injection. The latter, if combined with BARseq decoding, can provide an experimental prediction of the frequency of double-labeled events since connected-source networks are not labeled in the absence of RG.

We have added Table 2, which breaks down the 2,914 barcoded cells based on whether they are presynaptic or source cells, and which type of network they belong to. We agree with the reviewer that the additional Cre- or RG- control experiments in parallel would allow an independent estimate of the double labeled networks and the no-source networks. We have included added a discussion of possible controls to further optimize the trans-synaptic tracing approach in future studies in the Discussion.

Third, I would appreciate more quantitative data on the putative single-source network (Fig. 5I and S6) in terms of the distribution of pre- and post-synaptic TC cell types. The majority of labeling appeared to occur locally, with only two thalamic neurons observed in sample 25311842 (Fig. S6). How many instances of long-distance labeling (for example, > 500 microns away from the injection site) were observed in total? Is this low efficiency of long-distance labeling expected based on the utilized combinations of AAVs and RV vectors? A simple independent RV tracing solely detecting mCherry would be useful for evaluating the labeling efficiency of the method. I have experienced similar "less jump" RV tracing when RV particles were prepared in a single step, as this study did, rather than multiple rounds of amplification in traditional protocols, such as Osakada F et al Nat Protocol 2013.

We imaged an animal that was injected in parallel to assess labeling (now included in Supplementary Note 2 and Supp. Fig. S5). The labeling pattern in the newly imaged animal was largely consistent with the results from the barcoded experiment: most labeled neurons were seen in the vicinity of the injection site, and sparser labeling was seen in other cortical areas and the thalamus. We further found that most neurons that were labeled in the thalamus were about 1 mm posterior to the center of the injection site, and thus would not have been sequenced in the in situ sequencing experiment (in which we sequenced about 640 µm of tissue spanning the injection site).

In addition, we found that the bulk of the cells that expressed mCherry from the rabies virus only partially overlapped with the area that contained cells co-expressing BFP with the rabies glycoprotein. Moreover, very few cells co-expressed mCherry and BFP, which would be considered source cells in a conventional mono-synaptic tracing experiment. The small numbers of source cells likely also contributed to the sparseness of long-range labeling in the barcoded experiment.

These interpretations and comparisons to the barcoded experiment are now included in Supplementary Note 2.

**Reviewer #3 (Public Review):**
The manuscript by Zhang and colleagues attempts to combine genetically barcoded rabies viruses with spatial transcriptomics in order to genetically identify connected pairs. The major shortcoming with the application of a barcoded rabies virus, as reported by 2 groups prior, is that with the high dropout rate inherent in single cell procedures, it is difficult to definitively identify connected pairs. By combining the two methods, they are able to establish a platform for doing that, and provide insight into connectivity, as well as pros and cons of their method, which is well thought out and balanced.Overall the manuscript is well-done, but I have a few minor considerations about tone and accuracy of statements, as well as some limitations in how experiments were done. First, the idea of using rabies to obtain broader tropism than AAVs isn't really accurate - each virus has its own set of tropisms, and it isn't clear that rabies is broader (or can be made to be broader).

As the reviewer suggested, we toned down this claim and stated that rabies virus has different tropism to complement AAV.

Second, rabies does not label all neurons that project to a target site - it labels some fraction of them.

We meant to say that retrograde labeling is not restricted to labeling neurons from a certain brain region. We have clarified in the text.

Third, the high rate of rabies virus mutation should be considered - if it is, or is not a problem in detecting barcodes with high fidelity, this should be noted.

Our analysis showed that sequencing 15 bases was sufficient to tolerate a small number of mismatches in the barcode sequences and could distinguish real barcodes from random sequences (Fig. 4A). Thus, we can tolerate mutations in the barcode sequence. We have clarified this in the text.

Fourth, there are a number of implicit assumptions in this manuscript, not all of which are equally backed up by data. For example, it is not clear that all rabies virus transmission is synaptic specific; in fact, quite a few studies argue that it is not (e.g., detection of rabies transcripts in glial cells). Thus, arguments about lost-source networks and the idea that if a cell is lost from the network, that will stop synaptic transmission, is not clear. There is also the very real propensity that, the sicker a starter cell gets, the more non-specific spread of virus (e.g., via necrosis) occurs.

We agree with the reviewer that how strictly virus transmission is restricted to synapses remains a hotly debated question in the field, and this question is relevant not only to techniques based on barcoded rabies tracing, but to all trans-synaptic tracing experiments. A barcoding-based approach can generate single-cell data that enable direct comparison to other data modalities that measure synaptic connectivity, such as multi-patch and EM. These future experiments may provide additional insights into the questions that the reviewer raised. We have included additional discussion about how non-synaptic transmission of barcodes because of the necrosis of source cells may affect the analysis in the Discussion.

Regarding the scenario in which the source cell dies, we agree with the reviewer and have clarified in the revised manuscript.

Fifth, in the experiments performed in Figure 5, the authors used a FLEx-TVA expressed via aretrograde Cre, and followed this by injection of their rabies virus library. The issue here is that there will be many (potentially thousands) of local infection events near the injection site that TVA-mediated but are Cre-dependent (=off-target expression of TVA in the absence of Cre). This is a major confound in interpreting the labeling of these cells. They may express very low levels of TVA, but still have infection be mediated by TVA. The authors did not clearly explore how expression of TVA related to rabies virus infection of cells near the rabies injection site. A modified version of TVA, such as 66T, should have been used to mitigate this issue. Otherwise, it is impossible to determine connectivity locally. The authors do not go to great lengths to interpret the findings of these observations, so I am not sure this is a critical issue, but it should be pointed out by the authors as a caveat to their dataset.

We agree with the reviewer that this type of infection could potentially be a major contributor to no-source networks, which were abundant in our experiment. Because small no-source networks were excluded from our analyses, and large no-source networks were only included for barcodes with low frequency (i.e., it would be nearly impossible statistically to generate such large no-source networks from independent infections), we believe that the effect of independent infections on our analyses were minimized. We have added a control experiment in Fig S5 and Supplementary Note 2, which further supported the hypothesis that there were many independent infections. We also included additional discussion about how this can be assessed and optimized in future studies in the Discussion.

Sixth, the authors are making estimates of rabies spread by comparison to a set of experiments that was performed quite differently. In the two studies cited (Liu et al., done the standard way, and Wertz et al., tracing from a single cell), the authors were likely infecting with a rabies virus using a high multiplicity of infection, which likely yields higher rates of viral expression in these starter cells and higher levels of input labeling. However, in these experiments, the authors need to infect with a low MOI, and explicitly exclude cells with >1 barcode. Having only a single virion trigger infection of starter cells will likely reduce the #s of inputs relative to starter neurons. Thus, the stringent criteria for excluding small networks may not be entirely warranted. If the authors wish to only explore larger networks, this caveat should be explicitly noted.

In the trans-synaptic labeling experiment, we actually used high rabies titer (200 nL, 7.6e10 iu/mL) that was comparable to conventional rabies tracing experiments. We did not exclude cells with multiple barcodes (as opposed to barcodes in multiple source cells), because we could resolve multiple barcodes in the same cell and indeed found many cells with multiple barcodes. We have clarified this in the text.

Overall, if the caveats above are noted and more nuance is added to some of the interpretation and discussion of results, this would greatly help the manuscript, as readers will be looking to the authors as the authority on how to use this technology.

In addition to addressing the specific concerns of the reviewer as described above, we modified the Results and Discussion sections on the trans-synaptic tracing experiment to improve clarity to general readers and expanded the discussion on future optimizations.

**Reviewer #1 (Recommendations For The Authors):**
The scientific problem is clearly stated and well laid out, the data are clearly presented, and the experiments well justified and nicely discussed. It was overall a very enjoyable read. The figures are generally nice and clear, however, I find the legends excessively concise. A bit too often, they just sort of introduce the title of the panel rather than a proper explanation of what it is depicted. A clear case is for example visible in Fig 2, where the description of the panels is minimal, but this is a general trend of the manuscript. This makes the figures a bit hard to follow as self-contained entities, without having to continuously go back to the main text. I think this could be improved with longer and more helpful descriptions.

We have revised all figure legends to make them more descriptive.

Other minor things:In the cDNA synthesis step for in-situ sequencing, I believe the authors might have forgotten one detail: the addition of aminoallyl dUTP to the RT reaction. If I recall correctly this is done in BARseq. The fact that the authors crosslink with BS-PEG on day 2, makes me suspect they spike in these nucleotides during the RT but this is not specified in the relevant step. Perhaps this is a mistake that needs correction.

The RT primers we used have an amine group at 5’, which directly allows crosslinking. Thus, we did not need to spike in aminoallyl dUTP in the RT reaction. We have clarified this in the Methods.

**Reviewer #2 (Recommendations For The Authors):**
Throughout the manuscript, there are frequent references to the "Methods" section for important details. However, it can be challenging to determine which specific section of the Methods the authors are referring to, and in some cases, a thorough examination of the entire Methods section fails to locate the exact information needed to support the authors' claims. Below are a few specific examples of this issue. The authors are encouraged to be more precise in their references to the Methods section.

In the revised manuscript, we numbered each subsection of Methods and updated pointers and associated hyperlinks in the main text to the subsection numbers.

On page 7, line 14, it is unclear how the authors compared the cell marker gene expression with the marker gene expression in the reference cell type.

We have clarified in the revised manuscript.

On page 7, line 33, the authors note that some barcodes may have been missed during the sequencing of the rabies virus libraries, but the Methods section lacked a convincing explanation on this issue (see my point 2 above).

We included a separate subsection on the sequencing of rabies libraries and the analysis of the sequencing depth in the Methods. In this new subsection, we further clarified our reasoning for identifying the lack of sequencing depth as a reason for missing barcodes, especially in comparison to sequencing depth required for establishing exact molecule counts used in established MAPseq and BARseq techniques with Sindbis libraries.

On page 9, line 44, the authors state that they considered a barcode to be associated with a cell if they found at least six molecules of that barcode in a cell, as detailed in the Methods section. However, the rationale behind this level of stringency is not provided in the Methods.

We initially chose this threshold based on visual inspection of the sequencing images of the barcoded cells. Because the labeled cell types were consistent with our expectations (Fig. 4E-G), we did not further optimize the threshold for detecting retrogradely labeled barcoded cells.

I have noticed that some important explanations of figure panels are missing in the legends, making it challenging to understand the figures. Below are typical examples of this issue.

In addition to the examples that the reviewer mentioned below, we also revised many other figure panels to make them clear to the readers.

In Fig. 2, "RV into SC" in panel C does not make sense, as RV was injected into the thalamus. There is no explanation of the images in this panel C.

We have corrected the typo in the revision.

In Fig. 3, information on the endogenous gene panel for cell type classification (Table S3) could be mentioned in the legend or corresponding text.

We now cite Table S3 both in Fig 3 legend and in the main text. We also included a list of the 104 cell type marker genes we used in Table S3.

In panel J, it is unclear why the total number of BC cells is 2,752, and not 4,130 as mentioned in the text.

This is a typo. We have corrected this in the revision. The correct number (3,746) refers to the number of cells that did not belong to either of the two categories at the bottom of the panel, and not the total number of neurons. To make this clear, we now also include the total number of barcoded cells at the top of the panel.

In Fig. 4, the definitions of "+" and "−" symbols in panels K and L are unclear. Also, it seems that the second left column of panel K should read "T −."

We corrected the typo in K, further clarified the “Area” labels, and changed the “S” label in 4K to “−”. This change does not change the original meaning of the figure: when considering the variance explained in L4/5 IT neurons, considering the subclass compositional profile is equivalent to not using the compositional profiles of cell types, because L4/5 IT neurons all belong to the same subclass (L4/5 IT subclass). Although operationally we simply considered subclass-level compositional profiles when calculating the variance explained, we think that changing this to “−” is clearer for the readers.

In Fig. 5, panel E is uninterpretable.

We revised the main text and the figure to clarify how we manually proofread cells to determine the QC thresholds for barcoded cells. These plots showed a summary of the proofreading. We also revised the figures to indicate that they showed the fraction of barcoded cells that were considered real after proofreading. In the revised version, we moved these plots to Fig. S5.

In Fig. S1, I do not understand the identity of the six samples on the X-axis of panel A (given that only two animals were described in the main text) and what panel B shows, including the definition of map_cluster_conf and map_cluster_corr.

In the revised Fig. S1, we made it more explicit that the six animals include both animals used for retrograde tracing (2 animals) and those used for trans-synaptic tracing (4 animals). We updated the y axis labels to be more readable and cited the relevant Methods section for definitions.

In Fig. S2, please provide the definitions of blue and red dots and values in panel A, as well as the color codes and size of the circles in panel B. My overall impression from panel B is that there is no significant difference between RV-infected and non-infected cells. The authors should provide more quantitative and statistical support for the claim that "RV-infected cells had higher expression of immune response-related genes."

We toned down the statement to “Consistent with previous studies […], some immune response related genes were up-regulated in virus-infected cells compared to non-infected cells.” Because the main point of the single-cell RNAseq analysis was that rabies did not affect the ability to distinguish transcriptomic types, the change in immune response-related genes was not essential to the main conclusions. We clarified the red and blue dots in panel A and changed panel B to show the top up-regulated immune response-related genes in the revised manuscript.

In Fig. S3, the definitions of the color code and circle size are missing.

We have added the legends in Fig. S3.